# Exploring and applying the substrate promiscuity of a C-glycosyltransferase in the chemo-enzymatic synthesis of bioactive C-glycosides

Kebo Xie [1,2✉], Xiaolin Zhang[1,2], Songyang Sui[1], Fei Ye[1✉] & Jungui Dai [1✉]

Bioactive natural C-glycosides are rare and chemical C-glycosylation faces challenges while enzymatic C-glycosylation catalyzed by C-glycosyltransferases provides an alternative way. However, only a small number of C-glycosyltransferases have been found, and most of the discovered C-glycosyltransferases prefer to glycosylate phenols with an acyl side chain. Here, a promiscuous C-glycosyltransferase, AbCGT, which is capable of C-glycosylating scaffolds lacking acyl groups, is identified from *Aloe barbadensis*. Based on the substrate promiscuity of AbCGT, 16 C-glycosides with inhibitory activity against sodium-dependent glucose transporters 2 are chemo-enzymatically synthesized. The C-glycoside **46a** shows hypoglycemic activity in diabetic mice and is biosynthesized with a cumulative yield on the 3.95 g L$^{-1}$ scale. In addition, the key residues involved in the catalytic selectivity of AbCGT are explored. These findings suggest that AbCGT is a powerful tool in the synthesis of lead compounds for drug discovery and an example for engineering the catalytic selectivity of C-glycosyltransferases.

---

[1] State Key Laboratory of Bioactive Substance and Function of Natural Medicines, CAMS Key Laboratory of Enzyme and Biocatalysis of Natural Drugs, and NHC Key Laboratory of Biosynthesis of Natural Products, Institute of Materia Medica, Chinese Academy of Medical Sciences and Peking Union Medical College, 1 Xian Nong Tan Street, 100050 Beijing, China. [2]These authors contributed equally: Kebo Xie, Xiaolin Zhang. ✉email: keboxie@imm.ac.cn; yefei@imm.ac.cn; jgdai@imm.ac.cn

Sugar moieties attached to aglycons through O-, S-, N-, and C-glycosidic bonds are important for the pharmacological properties and physiological activities of many bioactive compounds[1–3]. Compared with O-, S-, N-glycosides, C-glycosides not only maintain the pharmacological properties but also are remarkably more stable, as their C–C bonds are resistant to glycosidase or acid hydrolysis[4–7]. The most widely known bioactive C-glycoside is dapagliflozin, the first global hypoglycemic drug as a sodium-dependent glucose cotransporter 2 (SGLT2) inhibitor. Dapagliflozin was developed from the natural O-glycoside phlorizin as the lead compound. Compared with phlorizin, the C-glycoside dapagliflozin exhibits not only higher inhibitory activity but also better selectivity toward SGLT2 and stability in vivo[8,9]. However, natural C-glycosides are comparatively rare, and chemical C-glycosylation is restricted by challenges, including poor regio- and stereoselectivity as well as the tedious protection and deprotection of functional groups[10–12]. Although a palladium-catalyzed C-glycosylation method was recently developed[13], the efficient and environment-friendly synthesis of bioactive C-glycosides remains challenging. By contrast, enzymatic C-glycosylation catalyzed by specific C-glycosyltransferases (CGTs) can alleviate these disadvantages and is considered to be a prospective approach to synthesize C-glycosides[14–16].

During the past decades, microbial CGTs have attracted considerable interest, and great progress has been achieved in identification of CGTs from bacteria[17–21]. However, studies on plant CGTs were rather limited until the year of 2009[22,23]. OsCGT was the first flavonoid CGT identified from Oryza sativa, which initiated the exploration of plant CGTs. From 2009 to 2020, about ten plant CGTs were identified[15,23–31]. Recently, a highly promiscuous CGT (GgCGT) with C-, O-, N-, and S-glycosylation activity was identified from Glycyrrhiza glabra[32]. GgCGT could efficiently catalyze C-glycosylation of substrates containing a flopropione unit. However, these discovered CGTs function primarily on C-glycosylating flavonoids[23–30] and other phenols with acyl groups[15,31,32]. In our previous work, although the substrate scopes of the mutated variants of a Mangifera indica CGT were expanded, the structural diversity of substrates is still limited, for example, an acyl substituent of phenolic acceptors is necessary for C-glycosylation[31]. Therefore, mining CGTs with catalytic stereospecificity and promiscuity toward other structurally different substrates and further establishing a green and economical biosynthesis route for bioactive C-glycosides are significant for drug design and discovery.

Aloe barbadensis is a traditional Chinese medicinal herb, and two types of bioactive C-glycosides, namely, aloin and aloesin (rare C-glycoside derivatives of anthraquinones and chromones, respectively), have been isolated from its leaves[33–35]. However, little is known about the corresponding CGTs, which prompted us to search for CGTs with specific catalytic properties. Herein, we report the identification of a CGT, designated AbCGT, from A. barbadensis. The substrate promiscuity and catalytic specificity of AbCGT were systematically investigated. Most importantly, the unusual catalytic property of AbCGT was further used to design and economically synthesize a series of unnatural C-glycosides with the aim of finding SGLT2 inhibitors.

## Results

### Cloning and identification of the C-glycosyltransferase gene.
The large amount of C-glycosides in Aloe barbadensis suggest the high expression level of the corresponding CGT genes. According to the transcriptome data, 30 candidate CGT genes (AbGT1–30) with high expression levels were selected. Subsequently, a neighbor-joining phylogenetic tree was constructed to analyze the evolutionary relationships of AbGT1–30 with GTs from different phyla (Supplementary Fig. 1). Intriguingly, AbGT27 and other plant CGTs were grouped into a single clade, which was clearly divergent from the OGTs. Thus, the full-length cDNA of AbGT27 was cloned and expressed in Transetta (DE3) Escherichia coli. The catalytic activity of AbGT27 was screened with phloracetophene (11) and phloretin (17), which are commonly accepted substrates for plant CGTs, and uridine 5′-diphosphoglucose (UDP-Glc). AbGT27 exhibited high glycosylation activity (conversion rates 100%) to phloracetophene (11) and phloretin (17) generating 11aa and 17aa, respectively (Supplementary Figs. 2–3). C-glycosylation can be distinguished from other types of glycosylation by MS/MS. The characteristic MS/MS fragment ions of C-glucosides are $[M-H-120]^−$ and $[M-H-90]^−$, while that of O-, S-, and N-glucoside is $[M-H-162]^−/[M+H-162]^+$[23,36,37]. Therefore, the fragment ions, $[M-H-240]^−$ and $[M-H-180]^−$, of 11aa and 17aa suggest the production of di-C-glucosides. AbGT27 was also able to catalyze mono-C-glucosylation. The ratio of mono- and di-C-glycosylated products was positively correlated with the molar ratio of the sugar acceptor and donor. If insufficient sugar donors (such as acceptor:donor = 1:0.5) were provided, only mono-C-glycoside (17a) was generated (Supplementary Fig. 4). However, when excess sugar donors (such as acceptor:donor = 1:2.5) were provided, the acceptor (17) was di-glycosylated, and only di-C-glycoside (17aa, in 100% yield) and its spontaneously oxidative derivative (17o) were generated. When the molar ratio of the sugar acceptor and donor was 1:1, both mono- and di-C-glycoside were observed with 52 and 22% yields, respectively. Therefore, mono- and di-C-glycosylated products of phloracetophene (11) and phloretin (17) were prepared with different ratios of substrates, and their structures were further identified as 3-C-β-D-glucosides (11a and 17a) and 3,5-di-C-β-D-glucosides (11aa and 17aa) by MS, $^1H$ and $^{13}C$ NMR, respectively. This catalytic property can be used to generate C-glycosides with one or two C-sugar moieties by adjusting the ratio of acceptor and donor substrates.

The cDNA sequence of AbGT27 (1413 bp, GenBank accession number MN747045) contained an ORF encoding 470 amino acids and showed the highest identity (56%) to an unidentified GT from Phoenix dactylifera[38]. AbGT27 was purified to >90% homogeneity, and its native function still remains unknown as its failure to accept hypothetical substrates (s1–s6) designed based on the proposed biosynthetic pathways of aloin and aloesin (Supplementary Figs. 5–6). Therefore, AbGT27 was tentatively assigned as a CGT designated AbCGT.

Generally, most of the known GTs, including OGTs and CGTs, exhibit the highest catalytic activity at pH 7.0–9.0 and lower catalytic activity at pH >10.0[15,23–32]. However, the maximum catalytic activity of AbCGT was observed at pH 11.0 (Supplementary Fig. 7a). C-glycosylation catalyzed by CGTs is achieved through direct nucleophilic displacement at the anomeric carbon by an aromatic carbanion, suggesting that the C-glycosylation is a Friedel–Crafts-like reaction[39]. Formation of the C–C-glycosidic bond is positively correlated with the electronic density of the aromatic carbon. At an alkaline environment (pH 11.0), the electronic density of the aromatic carbon is further improved due to the salification of aromatic hydroxyls. Therefore, with the increasing pH of the reaction buffer from 6.0 to 11.0, the yield of monoglycosylated product (17a) was firstly increasing and then decreasing due to the second glycosylation generating the diglycosylated product (17aa). This is the only CGT ever known with maximum catalytic activity in such an alkaline environment. For the effects of temperatures, the highest conversion rate was observed at 45 °C (Supplementary Fig. 7b). $Mn^{2+}$, $Mg^{2+}$, $Ca^{2+}$, and $Ba^{2+}$ greatly enhanced the catalytic activity of AbCGT (Supplementary Fig. 7c). This enhanced catalytic activity was

observed with short reaction time (10 min), as longer-time incubation resulted in the same conversion rates. The kinetic parameters of AbCGT with phloracetophene (**11**) and phloretin (**17**) were also determined (Supplementary Fig. 8). The high catalytic efficiency of AbCGT was represented by the high $K_{cat}$ values for phloracetophene (144 min$^{-1}$) and phloretin (120 min$^{-1}$).

**Investigation of the substrate promiscuity of AbCGT.** Plant CGTs usually recognize only substrates with the basic core structure of acyl phloroglucinol, which possesses three hydroxyl groups and one acyl group[15,23–26,29,31,32]. Thus, to explore the effects of the number and position of phenolic hydroxyl groups on the catalytic activity of AbCGT, an acceptor library containing acyl phenols (**1**–**3**), acyl resorcinols (**4**–**9**), acyl phloroglucinols (**10**–**19** and **23**), phloroglucinol derivatives (**20**–**22**) without basic acyl groups, and coumarins (**24**–**26**) was used for the reactions (Fig. 1). To probe the sugar tolerance of AbCGT, six different NDP sugars were also used as sugar donors (Fig. 1).

From the first-pass analysis with HPLC-UV/ESIMS, AbCGT exhibited *O*-glycosylation activity to acyl phenols with one hydroxyl (**1**–**3**), *C*-glycosylation activity to acyl phloroglucinol with three hydroxyls (**11**) and mixed *O*-/*C*-glycosylation activities to acyl resorcinols with two hydroxyls (**5**–**7**) (Fig. 1 and Supplementary Figs. 9–14). Similarly, this change in catalytic reaction patterns depending on the hydroxyl groups was also observed with acceptors **8**–**10**, **12**–**19**, and **23** (Supplementary Figs. 15–24). As mentioned above, the formation of the C–C-glycosidic bond is positively correlated with the electronic density of the aromatic carbon, which can be increased by the number of substituted hydroxyl(s). Therefore, the *O*-glycosylation activity (acyl phenols) was changed to *C*-glycosylation activity (acyl phloroglucinols) through the stage of *O*-/*C*-glycosylation activity (acyl resorcinols). For 2,4-dihydroxyacetophenone (**6**) with two phenolic hydroxyl groups, AbCGT showed both *C*- and *O*-glycosylation activity simultaneously. However, when the H-6 of **6** was replaced by a methyl group (electron-donating group) (**9**), AbCGT showed specific *C*-glycosylation activity with the high conversion rate. The methyl substitution may increase the electron density of the aromatic ring for nucleophilic attack in *C*-glycosidic bond formation. Logically, the *O*-methylation of phenolic hydroxyl groups (**13**–**15**) decreased the glycosylation activity of AbCGT as the electron-donating ability of the hydroxymethyl was weaker than that of the hydroxyl. According to the structures of the substrates and isolated products, the *C*-glycosylation catalyzed by AbCGT needs at least two hydroxyl groups in the aromatic rings of acceptors and occurs in the *ortho*-position of one phenolic hydroxy. AbCGT seems to have no regioselectivity in *O*-glycosylation of the small molecules (**1**–**3**). The *O*-glycosylation can occur in any hydroxyl groups of the aromatic ring.

Rare di-*C*-glycosides generally possess various bioactivities, and some have been used as clinical drugs in China, such as hydroxysafflor yellow A[40]. When excess sugar donors were provided, AbCGT showed high di-*C*-glycosylating activity with conversion rates of 100% to the acceptors (**10**, **11**, and **16**–**19**) with the unit of phloroglucinol. Therefore, AbCGT is a prospective enzymatic tool for the synthesis of rare bioactive di-*C*-glycosides.

In addition, according to the LCMS data, AbCGT catalyzes 7-hydroxy-4-methylcoumarin (**24**), 7-mercapto-4-methylcoumarin (**25**) and 7-amino-4-methylcoumarin (**26**) to generate their corresponding *O*-, *S*-, and *N*-glucosides, indicating that AbCGT is a versatile GT with *C*-, *O*-, *N*-, and *S*-glycosylation activities (Supplementary Figs. 25–27). Besides the aglycon promiscuity, AbCGT also possesses a broad substrate spectrum with diverse

sugar donors. When phloretin (**17**) was used as an acceptor, AbCGT accepted five sugar donors, including UDP-α-D-glucose, TDP-α-D-glucose, UDP-α-D-galactose, UDP-α-D-xylose, and UDP-α-D-glucuronic acid, to generate the corresponding *C*-glycosides, respectively (Supplementary Figs. 28–32 and Supplementary Table 1). Therefore, besides the *C*-glucosides (**17a** and **17aa**), the *C*-galactoside (**17b**) and *C*-xyloside (**17c**) were also prepared and their structures were identified as phloretin-3′-*C*-β-D-galactoside (**17b**) and phloretin-3′-*C*-β-D-xyloside (**17c**) by MS, $^1$H and $^{13}$C NMR, respectively. Biological assays revealed that *C*-glucoside **17a** with only one glucosyl moiety exhibited higher SGLT2 inhibitory activity than those of other *C*-glycosides (**17aa**, **17b**, and **17c**) (Supplementary Table 2). Therefore, UDP-Glc is selected as the sugar donor in the following application of AbCGT in the synthesis of *C*-glycosides.

Acyl groups, as electron-donating and potential positioning groups, are necessary for the substrates of known plant CGTs[15,23–32]. Among the substrates of acyl resorcinols, which should be both *C*- and *O*-glycosylated according to the above catalytic characteristics of AbCGT, there was an exception that 3,5-dihydroxyacetophenone (**4**) was only *O*-glycosylated (Supplementary Fig. 33). In comparison to other acyl resorcinols (**5**–**9**), the acyl group of 3,5-dihydroxyacetophenone (**4**) is not in *ortho*-position to the hydroxyl groups. Therefore, we initially believed that the *C*-glycosylation activity of AbCGT was also dependent on the acyl group, similar to other known CGTs. However, to our surprise, AbCGT exhibited catalytic activity when phloroglucinol derivatives (**20**–**22**) without acyl groups attached to the aromatic rings were used as substrates (Supplementary Figs. 34–36). To further confirm the enzymatic reactions, the unnatural products of aglycons **20** and **21** were isolated from scaled-up reactions and characterized as 3-*C*-β-D-glucosides (**20a** and **21a**) by MS, $^1$H and $^{13}$C NMR. AbCGT exhibited 100% conversion rates to **20** and **21**, suggesting its substrate tolerance to unusual substrates.

**Design and application of AbCGT in chemoenzymatic synthesis of potential SGLT2 inhibitors.** The *O*-glycoside phlorizin is a SGLT1/2 inhibitor[41]. However, due to the poor selectivity for SGLT2, metabolic instability and substantial side effects, phlorizin failed in the clinical treatment of diabetes. Based on the prototypic phlorizin, various SGLT2 inhibitors have been developed by changing the types of glycosidic bonds and replacing the acyl groups with saturated carbon atoms[42]. In addition, biological assays revealed that products **20a** and **21a** without acyl groups displayed inhibitory activity toward SGLT2 (Table 1). Therefore, to explore the catalytic activity of AbCGT in recognizing unusual substrates without acyl groups and to further apply AbCGT in antidiabetic drug discovery, diverse aglycons of potential SGLT2 inhibitors were designed (Fig. 2a). Firstly, three commercially available *C*-/*N*-substituted resorcinols (**27**–**29**) were prepared. Then we designed and chemically synthesized 25 potential substrates (**30**–**54**). Functional groups, including aliphatic carbon chains (**30**–**33**), saturated carbon cycles (**34**–**37**), carbon chain-substituted benzenes (**38**–**41**), and *N*- and *S*-heterocycles (**42**–**45**), were chemically attached to phloroglucinol by a one-step reaction[43]. Furthermore, various derivatives (**46**–**54**) of 1-benzylbenzene-2,4,6-triol (**38**) and 1-phenethylbenzene-2,4,6-triol (**39**) were chemically synthesized to investigate the effects of different substituents in the B ring on the pharmacological activity. The introduction of sugar moieties to these acceptors was performed with purified AbCGT and whole-cell system, respectively (Fig. 2b, c).

The substrate tolerance and catalytic efficiency of purified AbCGT were emerged by the capability of recognizing all types of chemically synthesized unusual substrates, and 85% (24 of 28) of the substrates were completely *C*-glycosylated to their

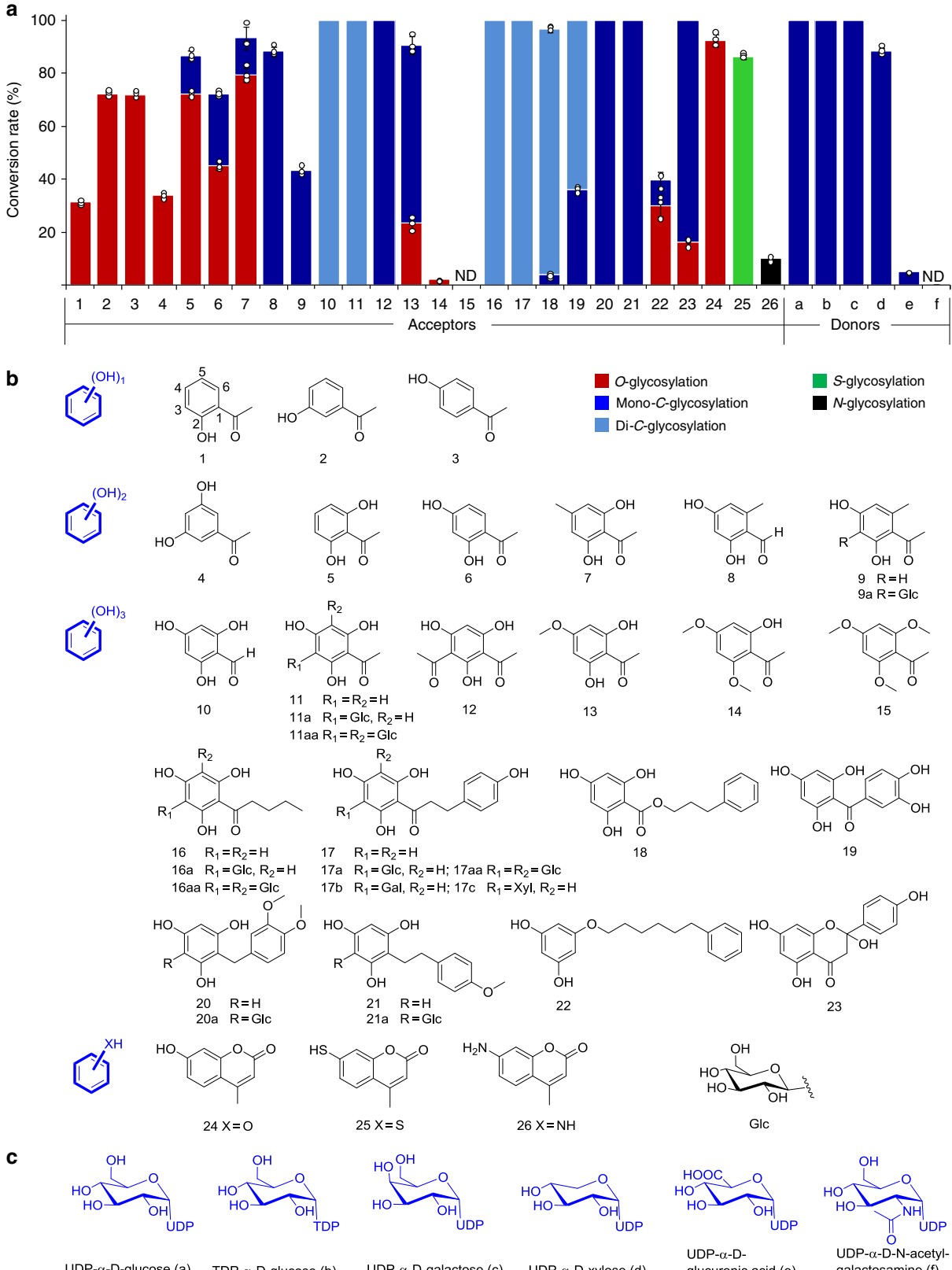

**Fig. 1 Probing the aglycon acceptor and sugar donor promiscuity of AbCGT. a** Percent conversion of each aglycon acceptor with AbCGT. The aglycon acceptors and sugar donors are listed based on the structural scaffolds shown in part **b** and **c**. The red, blue, light blue, green, and black columns represent conversion rates of *O*-, mono-*C*-, di-*C*-, *S*-, and *N*-glycosylation reactions, respectively. **b** The structures of the aglycon acceptors and corresponding glycosylated products prepared from the scale-up enzymatic reactions. **c** The structures of sugar donors used in this work. The sugar donor promiscuity was tested with phloretin (**17**) as the acceptor. ND not detected. Experiments were performed at pH 7.4 and 30 °C for up to 12 h. Conversion rates represent mean ± SD of three independent replicates ($n = 3$).

corresponding *C*-glycosides (Fig. 2b and Supplementary Figs. 37–55). For the acceptors with a basic unit of phloroglucinol, AbCGT exhibited 100% conversion rates, except for aglycon **45** with a bulky substituent group. Reasonably, AbCGT can also *C*-glycosylate all derivatives (**46**–**54**) of 1-benzylbenzene-2,4,6-triol

(**38**) and 1-phenethylbenzene-2,4,6-triol (**39**) with 100% conversion rates. To the best of our knowledge, no known CGT has such high efficiency to acceptors without acyl groups. Subsequently, scaled-up reactions were performed, and 19 enzymatic *C*-glycosides (**20a**, **21a**, **28a**, **30a**, **38a**, **39a**, **42a**–**44a**, **46a**–**54a**, and **47aa**) were prepared. The structures, including the position of the introduced sugar moiety and the anomeric stereochemistry, were elucidated by MS, [1]H and [13]C NMR spectroscopic analyses. All *C*-glycosylation sites of the enzymatic products were regioselective at C-3/3′. Large anomeric proton-coupling constants indicated the stereo-specific formation of β-anomers and an inversion mechanism for AbCGT, as UDP-Glc possesses an α-anomer (Supplementary Table 3). Among the 19 prepared *C*-glycosides, 18 (**20a**, **21a**, **28a**, **38a**, **39a**, **42a**–**44a**, **46a**–**54a**, and **47aa**) were not reported previously. Therefore, potential SGLT2 inhibitors were redesigned and efficiently synthesized by applying AbCGT in a chemoenzymatic method.

**Table 1 Inhibition and selectivity of representative *C*-glycosides on SGLT2.**

| *C*-glycosides | SGLT2 inhibitory rates[a] (%) | IC$_{50}$ (M) | SGLT1 inhibitory rates[a] (%) | IC$_{50}$ (M) |
|---|---|---|---|---|
| 46a | 100 | $1.74 \times 10^{-7}$ | 50.4 | $\approx 1 \times 10^{-5}$ |
| 47a | 100 | $2.09 \times 10^{-7}$ | 17.1 | $>1 \times 10^{-5}$ |
| 49a | 98.5 | $4.49 \times 10^{-7}$ | 34.7 | $>1 \times 10^{-5}$ |
| 44a | 96.6 | $4.21 \times 10^{-7}$ | 31.8 | $>1 \times 10^{-5}$ |
| 50a | 96.3 | $4.89 \times 10^{-7}$ | 27.1 | $>1 \times 10^{-5}$ |
| 48a | 85.4 | \ | \ | \ |
| 38a | 76.4 | $3.09 \times 10^{-6}$ | \ | \ |
| 42a | 68.5 | \ | \ | \ |
| 21a | 22.7 | \ | \ | \ |
| 20a | 13.6 | \ | \ | \ |
| Dapagliflozin | 102.4 | $2.24 \times 10^{-10}$ | 98.4 | $6.20 \times 10^{-7}$ |

\ not detected.
[a]The inhibitory rates were obtained with *C*-glycosides at 10 μM. Experiments were performed in triplicate.

**Evaluation of the SGLT2 inhibitory activity of *C*-glycosides in vitro.** The SGLT2 inhibitory activity of the 16 designed *C*-glycosides (**20a**, **21a**, **38a**, **39a**, **42a**–**44a**, and **46a**–**54a**) and their 16 corresponding aglycons was firstly evaluated at the cellular level. All *C*-glycosides exhibited inhibitory activity against SGLT2 (Fig. 3). Furthermore, five (**44a**, **46a**, **47a**, **49a**, and **50a**) of these designed *C*-glycosides showed inhibitory rates >96% at a concentration of $10^{-5}$ M, and their IC$_{50}$ values against SGLT2 were $1.74–4.89 \times 10^{-7}$ M (Table 1). Notably, the five *C*-glycosides also exhibited high selectivity in inhibiting SGLT2, as their IC$_{50}$ values

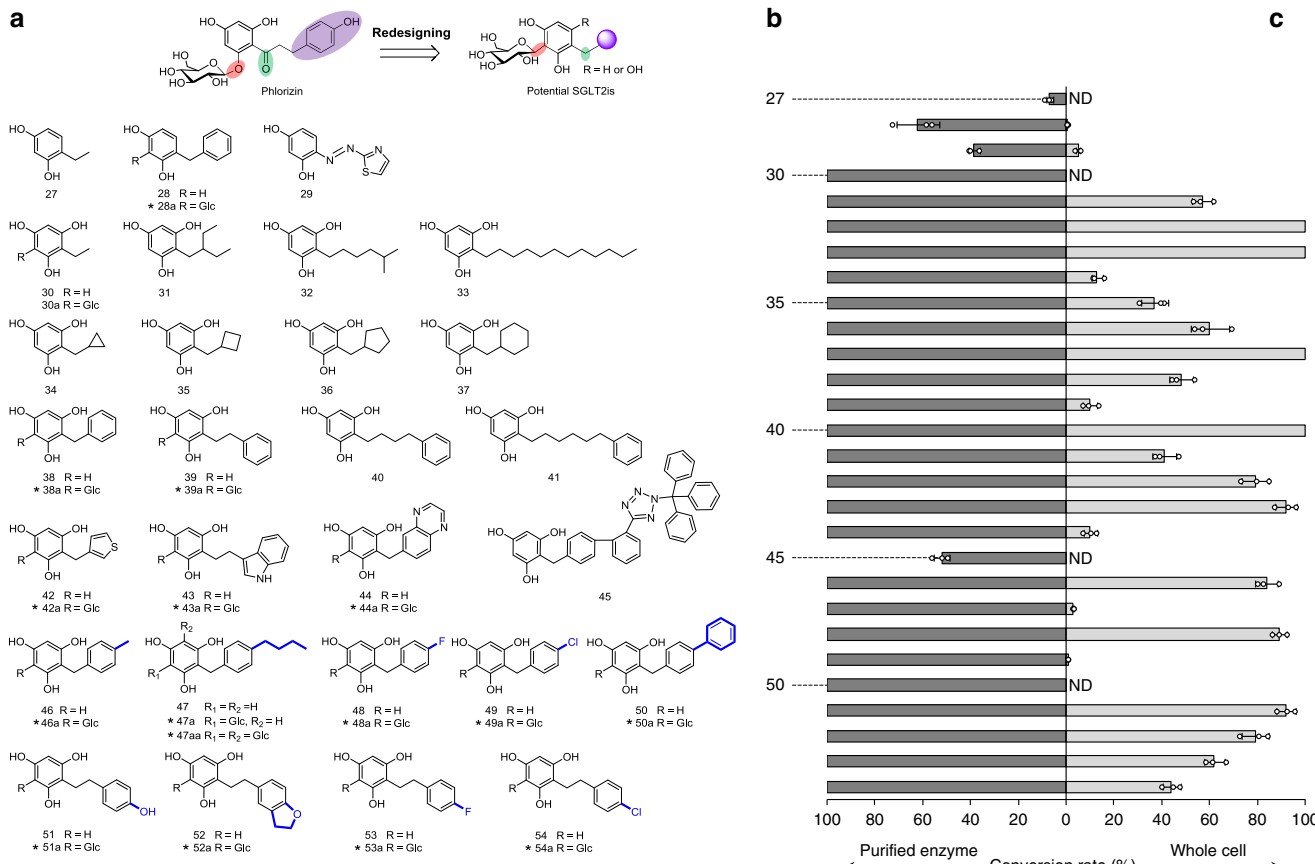

**Fig. 2 Establishing a chem-biosynthesis method for potential SGLT2 inhibitors. a** The redesigned structures of the substrates and corresponding glycosylated products. **b** Conversion rate of each substrate with purified AbCGT. **c** Conversion rate of each substrate with whole-cell system. *Not reported *C*-glycosides previously. ND not detected. Experiments were performed at 30 °C in M9 medium for 12 h with shaking (200 rpm), and the conversion rates represent mean ± SD of three independent replicates (*n* = 3).

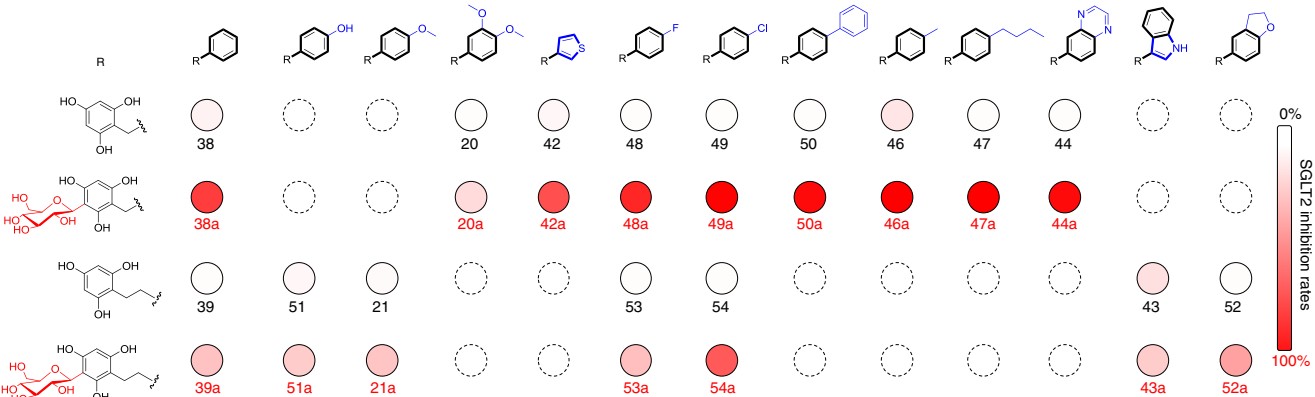

**Fig. 3 SGLT2 inhibition rates of the designed aglycons and the corresponding C-glycosides.** The shade of red filled circles represents the SGLT2 inhibition level with aglycons and C-glycosides at 10 μM. The depth of red color represents the SGLT2 inhibition rate. Experiments were performed in triplicate.

against SGLT1 were much higher (≥$10^{-5}$ M) than those against SGLT2 (~$10^{-7}$ M). It is worth mentioning that all the aglycons exhibited no inhibitory activity or much lower inhibitory activity than the corresponding C-glycosides (Fig. 3). Therefore, the C-linked sugar moiety not only generated structurally different compounds but also afforded aglycons with SGLT2 inhibitory activity and selectivity. The C-glycosylated products (**38a** and **46a–50a**) of 1-benzylbenzene-2,4,6-triols showed higher SGLT2 inhibitory activity than those (**39a** and **51a–54a**) of 1-phenethylbenzene-2,4,6-triols. These findings demonstrated that the benzyl group is a better substituent than the ethylbenzene group for bioactivity. Different substituents in the aromatic ring of the benzyl group (B ring) also resulted in different degrees of pharmacological activity. Replacing the B ring of **38a** with a thiophene group decreased the SGLT2 inhibitory activity, while halogen, phenyl, and alkyl group substitution enhanced the inhibitory activity (Fig. 3). The methyl substitution in the B ring (**46a**, IC$_{50}$ $1.74 \times 10^{-7}$ M) increased the inhibitory activity more than 17 times compared to that of the unsubstituted glycoside (**38a**, IC$_{50}$ $3.09 \times 10^{-6}$ M).

**Establishment, optimization, and utilization of a biocatalytic system for the production of SGLT2 inhibitors.** The potent SGLT2 inhibitory activity of C-glycoside **46a** inspired us to further evaluate in vivo activity as the candidate drug hit. Although AbCGT efficiently synthesized bioactive C-glycosides with UDP-Glc (~400 USD per gram) as the sugar donor at a laboratory scale, for large-scale preparation, the high cost of the sugar donors and the instability of the catalyst greatly limit the practical application of GTs. Thus, an engineered E. coli strain harboring its native biosynthetic pathway of UDP-Glc and pET28a-AbCGT was developed as a catalyst. As shown in Fig. 2c, this engineered strain economically glycosylated most of the potential acceptors without adding expensive UDP-Glc. Therefore, a chemo-biosynthetic approach combining a simple one-step chemical method for the synthesis of aglycons and an efficient biocatalytic method for C-glycosylation was successfully established for the total synthesis of **46a**.

To economically obtain **46a** with yield in grams scale, the cell density of cultures and input concentration of substrate **46** of the biocatalytic system were optimized. At a cell density of OD$_{600}$ = 13.0, the whole-cell biocatalyst glycosylated 300 mg L$^{-1}$ of acceptor **46** to produce 450 mg L$^{-1}$ of C-glycoside **46a** with a conversion rate of 90% (Fig. 4a, b). Aglycon **46** was chemically synthesized by a one-step method with an isolated yield of 65%, and the regio- and stereoselective introduction of the C-sugar

moiety was accomplished by a biocatalyst with an isolated yield of 80%. Thus, the total synthesis of SGLT2 inhibitor **46a** was accomplished with the isolated yield of 52%. Generally, in vitro enzymes are often unstable as one-time catalysts. In the whole-cell system, however, the AbCGT enzyme is located inside the cell, which provides a relatively stable environment. Thus, the reactions were scaled up to 500 mL to investigate the recyclability and stability of the whole-cell biocatalyst. The biocatalyst was reused more than ten times, which means one batch of whole-cell catalyst could catalyze more than ten batches of acceptor **46** (Fig. 4c). Cumulatively, 3.95 g L$^{-1}$ of target C-glycoside **46a** was generated by whole cells with an average conversion rate of 75%. The recyclability and stability indicated E. coli-AbCGT is a potential biocatalyst for the synthesis of C-glycosides for industrial applications.

**Evaluation of the pharmacological activity of C-glycosides in vivo.** With sufficient amount of **46a** in hand, its hypoglycemic activity was evaluated in three mouse models, including normal ICR mice, alloxan-induced diabetic mellitus (DM) mice, and T2DM KKAy mice. The oral glucose-tolerance test (OGTT) showed that 100 mg kg$^{-1}$ **46a** significantly lowered the peak value of blood glucose after glucose loading in ICR mice (Fig. 5a, b). In alloxan-induced DM mice, **46a** exhibited obviously hypoglycemic activity 0 − 8 h after oral administration (Fig. 5c, d). In KKAy mice, the blood glucose level was decreased at 1 h after **46a** administration and was maintained at a lower level for 24 h compared to the control (solvent-treated) group (Fig. 5e, f). Given the SGLT2 inhibitory activity of **46a** in vitro, the hypoglycemic mechanism may have involved the prevention of renal glucose reabsorption with elevated urinary glucose excretion (Fig. 5g). These hypoglycemic activities suggested that C-glycoside **46a** is a promising drug lead for treating diabetes.

**Exploration of the active sites involved in the C-/O-glycosylation activity.** CGTs and OGTs share similar structures but different functions[30]. The related key residues determining reaction patterns are still unknown. Finding the switch that controls the C- and O-glycosylation activity of GTs is significant to enzymatic glycosylation mechanism and protein engineering. As mentioned above, AbCGT showed both C- and O-glycosylation activity toward 2-hydroxynaringenin (**23**) (Fig. 1 and Supplementary Fig. 24), suggesting two different relative positions of substrates in the binding pocket, that provided an example to explore the key amino acids residues involved in C- and O-glycosylation. Homology modeling was performed with the template GgCGT

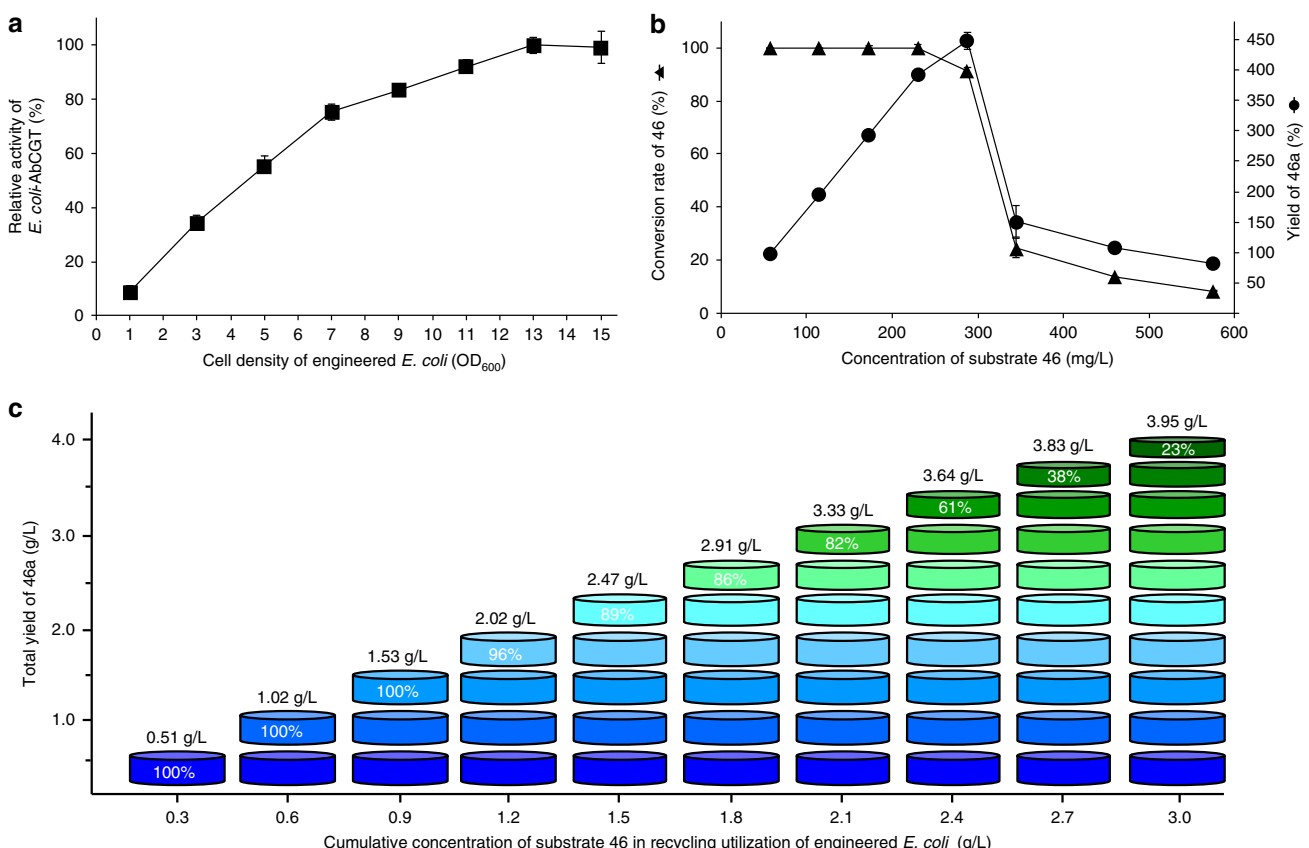

**Fig. 4 Optimizing the cell density and substrate concentration and exploring the recyclability and stability of the biocatalytic system. a** The effects of cell density to the catalytic activity of *E. coli-AbCGT*. **b** The effects of substrate concentration to the conversion rate of aglycon **46** and production of *C*-glycoside **46a**. **c** Recycling utilization of engineered cells with aglycon **46** (final concentration 0.3 g/L/round) for ten rounds in production of *C*-glycoside **46a**. Reactions were performed at 25 °C in M9 medium for 12 h with shaking (180 rpm). The total yield of **46a** and the conversion rate of **46** for each round are indicated. The relative activity (**a**) and conversion rates (**b**) represent mean ± SD of three independent replicates ($n = 3$).

(PDB entry, 6L5R.1), a di-*C*-glycosyltransferase[32]. It is reported that Val200 of UGT85H2, an (iso)flavonoid UGT from *Medicago truncatula*, is interacting with the 4′-hydroxyl group of the substrate kaempferol[44,45]. Mutating this neutral residue to that with charged polar side chain could greatly influence the enzyme activity[45]. A similar key amino acid residues Val183, which interacts with the 4′-hydroxyl group of 2-hydroxynaringenin (**23**), was found in AbCGT based on molecular docking and sequence alignment (Fig. 6a, b). Therefore, saturation mutagenesis was performed on Val183 of AbCGT.

When Val183 was mutated to residues of Asp, Glu and Pro, respectively, the *C*-glycosylation selectivity was greatly improved with 2-hydroxynaringenin (**23**) (Fig. 6c and Supplementary Figs. 56–57). Furthermore, the *O*-glycosylation activity of these variants was decreased with all acceptors (**1**–**7**, **13**, **14**, **22**, and **23**) those can be *O*- or both *C*- and *O*-glycosylated by wild-type AbCGT (Fig. 6d). The affinity of the variants for 2-hydroxynaringenin (**23**) is higher than that of wild-type AbCGT, while the enzyme efficiency ($K_{cat}/K_m$) of the variants is lower than that of wild-type AbCGT (Supplementary Fig. 58). It seems that the mutations (V183D V183E and V183P) in AbCGT affected both the catalytic selectivity and efficiency of the enzyme. The V183D and V183E variants showed relatively high specific *C*-glycosylation activity to 2-hydroxynaringenin (**23**). The potential mechanism might be the binding sites of AbCGT not tightly capturing the acceptor, thereby resulting in flexible substrate orientation. Thus, both *O*- and *C*-glycosylation sites of 2-hydroxynaringenin (**23**) could be positioning to UDP-Glc. The mutant residue with a charged polar side chain may form a

stronger interaction with the acceptor, which further fixes the orientation of the acceptor[45]. However, this hypothesis cannot explain the enhanced catalytic selectivity of the V183D and V183E variants to smaller acceptors without the 4′-hydroxyl group. The work on the structural biology of AbCGT is in progress and will help to elucidate the catalytic mechanism. Even so, investigation of the key amino acid residues of AbCGT involved in the control of *C*-/*O*-glycosylation provides hints for the further engineering of other promiscuous CGTs and insights into the switch between the *C*- and *O*-glycosylation activities of GTs.

## Discussion

In this study, we explored and applied the substrate promiscuity of AbCGT, a CGT from *A. barbadensis*. As a natural plant CGT, AbCGT exhibited amazing catalytic activity and promiscuity toward unnatural acceptors, especially toward those without acyl side chains which are usually good potential SGLT2 inhibitors. Chemical synthesis of this type of *C*-glycosides including clinical gliflozins, faces challenges involving the protection and deprotection of hydroxyls as well as poor stereoselectivity and complex synthetic routes. To overcome these obstacles, AbCGT was developed as a tool for stereoselective *C*-glycosylation with high efficiency. Combining enzymatic *C*-glycosylation with the chemical approach for synthesizing aglycons, bioactive *C*-glycosides were totally synthesized by two simple steps, and a chemo-biocatalysis route for the synthesis of *C*-glycosides was constructed. In this route, common glucose was used as the sugar

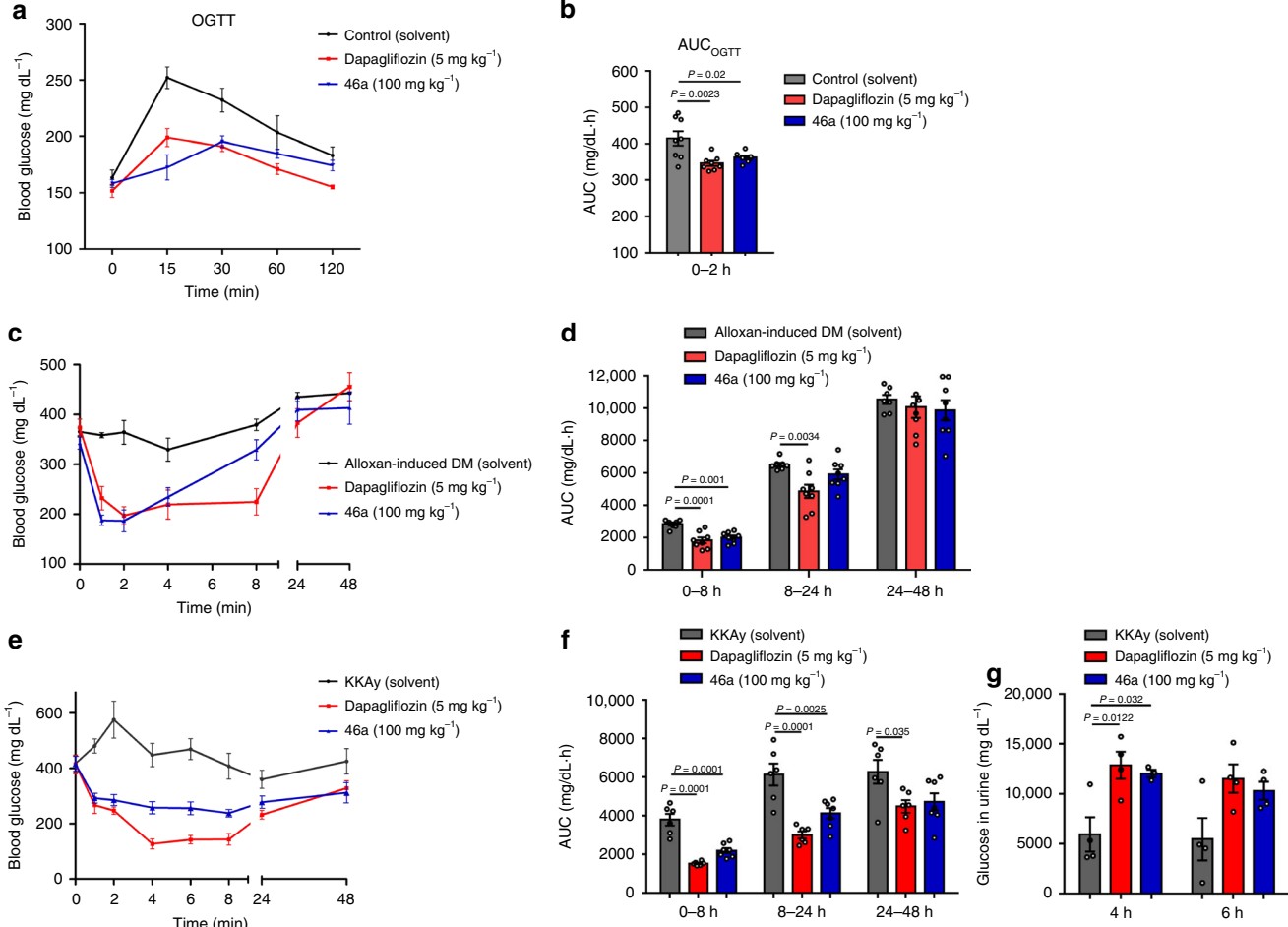

**Fig. 5 Evaluation of SGLT2 inhibitory activity of C-glycoside 46a in vivo. a**, **b** show **46a** enhanced glucose tolerance in ICR mice. **a** Summary data for changes of blood glucose after glucose loading in control ($n = 8$). Dapagliflozin-treated ($n = 8$), and **46a**-treated ($n = 7$) mice in OGTT. **b** Area under the blood glucose-time curve (AUC) in OGTT. **c**, **d** indicate the hypoglycemic effects of **46a** (single administration) in alloxan-induced diabetic mellitus (DM) mice. **c** Summary data for changes in blood glucose in DM ($n = 8$). Dapagliflozin-treated ($n = 8$), and **46a**-treated ($n = 8$) mice; **d** AUC for 0–8 h, 8–24 h, 24–48 h, respectively. **e**– **g** exhibit **46a** (single administration) promoted urinary glucose excretion and lowered the blood glucose in T2DM KKAy mice. **e** Summary data for changes of blood glucose in KKAy ($n = 6$). Dapagliflozin-treated ($n = 7$), and **46a**-treated ($n = 7$) mice; (**f**), AUC for 0–8 h, 8–24 h, 24–48 h, respectively; **g** glucose in urine at 4 h and 6 h ($n = 4$). The solvent was used as the blank control, and dapagliflozin was used as positive control. Data are represented as mean ± SEM. Significance was determined using one-way ANOVA with Dunnett's multiple comparisons test. Exact $P$ values ($P < 0.05$) are provided in the corresponding figures, respectively.

donor for the second step, and whole-cell biocatalyst can be reused for times making the cost greatly decreased and lending potential to this route for industrial application. Based on this route, diverse bioactive C-glycosides were designed and chemoenzymatically synthesized. This type of C-glycosides possessed SGLT2 inhibitory activity and showed a significant therapeutic effect on diabetic mice. In addition, some SGLT2 inhibitors such as **44a** even exhibited anti-influenza A (H1N1) activity (IC$_{50}$ ~10$^{-6}$ M, Supplementary Table 4), which suggested the wide application of AbCGT in the synthesis of bioactive glycosides other than SGLT2 inhibitors. In this study, the natural AbCGT breaks through the dependency of the acyl group in CGT catalyzing C-glycosylation, and diverse SGLT2 inhibitors were successfully chem-biosynthesized.

AbCGT is a versatile GT that simultaneously possesses C-, O-, S-, and N-glycosylation activities. Increasing numbers of CGTs have been found to possess multifunctional characteristics in forming glycosidic bonds. For examples, UGT708A6 from *Zea mays* exhibits both C- and O-glycosylation activities[24], MiCGT from *M. indica* showed C-, O-, and N-glycosylation activities[15],

and GgCGT from *G. glabra* generates C-, O-, N-, and S-glycosides[32]. The multifunctional characteristics hint the GTs catalyzing C-, O-, N-, and S-glycosylation is determined by substrate structures and enzyme-binding sites. Furthermore, the recently reported crystal structures of TcCGT1 and GgCGT, which are similar to the structures of OGTs, also supports this hypothesis[30,32]. The catalytic types may be determined by several key active sites. For multifunctional AbCGT, one possible binding site involved in C- and O-glycosylation was identified, and the C-glycosylation specificity of AbCGT was improved by rational design. In addition, progress in the engineering of AbCGT to improve the C-glycosylation specificity on aromatic aglycons with two hydroxyls was achieved in this work, and the door of enhancing the C-glycosylation activity of CGTs was opened. These findings not only provide a way to artificially manipulate the substrate promiscuity of CGTs but also shed light on the mechanisms of C- and O-glycosylation. Further studies on screening or engineering CGTs to recognize aromatic aglycons with only one hydroxyl or even no hydroxyl will be very challenging but significant for

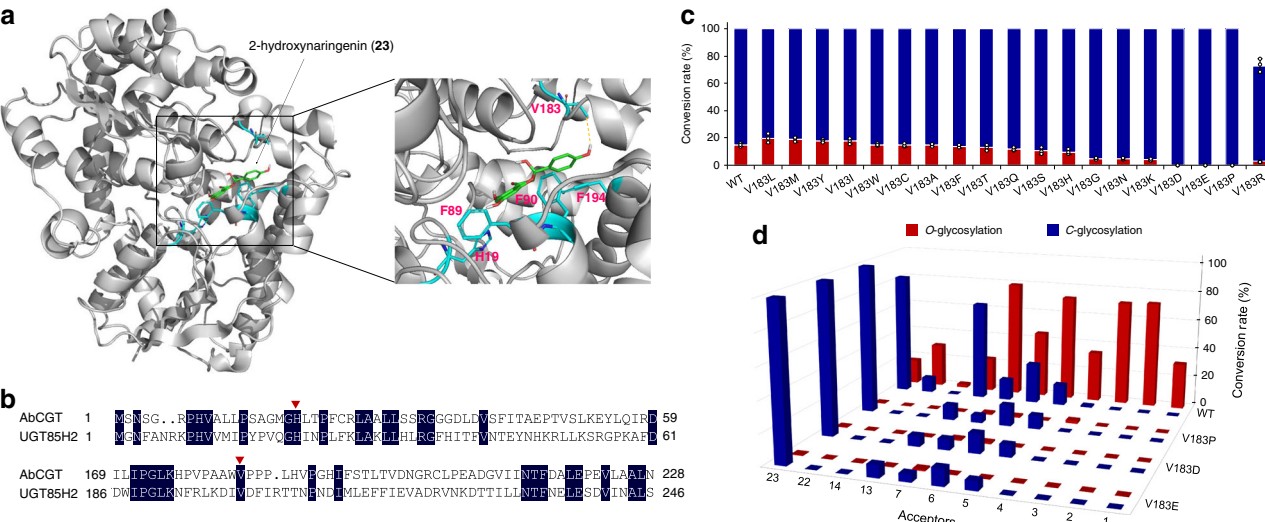

**Fig. 6 Engineering the catalytic specificity of AbCGT in C-/O-glycosylation. a** Structure homology modeling of AbCGT. The crystal structure of GgCGT (PDB entry, 6L5R.1) was used as a template for homology modeling. 2-hydroxynaringenin was docked into the putative binding pocket of AbCGT. H19, F89, F90, F194, and V183 are labeled. **b** Sequence alignment of AbCGT and UGT85H2. The conversed amino acid residues (H19 and V183) in the binding pocket are labeled with red triangles. **c** Percent conversion of 2-hydroxynaringenin by AbCGT and AbCGT variants. **d** Exploring the catalytic specificity of variants V183D, V183E and V183P in C-/O-glycosylation. The red, blue columns represent conversion rates of O- and C-glycosylation reactions, respectively. Reactions were performed at pH 7.4 and 30 °C for 12 h. The conversion rates represent mean ± SD of three independent replicates (n = 3).

CGTs to progress from the laboratory to industry for the synthesis of clinical C-glycoside drugs and biological C-glycosides.

## Methods

**General**. Chemicals and reagents including commercially available aglycons (**1–15**, **17**, **19**, and **23–29**) and sugar donors were purchased from Sigma-Aldrich (St. Louis, USA), J&K Scientific Ltd (Beijing, China), InnoChem Science & Technology Co., Ltd (Beijing, China), BioBioPha (Kunming, China), Beijing BG Biotech Co., Ltd (Beijing, China), and Beijing Adhoc International Technologies Co., Ltd (Beijing, China). Aglycons **16**, **18**, and **30** were chemically synthesized in our previous work[15], and aglycons **20–22** and **31–54** were synthesized in this work. KOD-Plus neo DNA polymerase was purchased from Toyobo Biotech Co., Ltd. (Shanghai, China). Primers were synthesized, and DNA was sequenced by Sangon Biotech Co., Ltd. (Shanghai, China). Restriction enzymes were purchased from Takara Biotechnology Co., Ltd. (Dalian, China). The analyses of substrate specificity and the determinations of the conversion rates were performed on an Agilent 1260 series HPLC system (Agilent Technologies, Germany) coupled with an LCQ Fleet ion trap mass spectrometer (Thermo Electron Corp., USA) equipped with an electrospray ionization (ESI) source. For quantification, three parallel assays were routinely performed. HRESIMS data were measured with an Agilent Technologies 6520 Accurate Mass Q-TOF LC/MS spectrometer. The $^1H$ and $^{13}C$ NMR data of prepared aglycons and C-glycosylated products were obtained by Bruker AVIIIHD-600 spectrometer.

**Plant materials**. Fresh leaves of *Aloe barbadensis* were collected, immediately frozen with liquid nitrogen, and stored at −80 °C until use.

**Molecular cloning of *AbGTs***. The total RNA of *A. barbadensis* leaves was prepared using the E.Z.N.A.™ Plant RNA Kit (Omega Bio-Tek, USA) and reverse-transcribed to cDNA with SmartScribe reverse transcriptase (Clontech, USA) following the manufacturer's instructions. Full-length cDNAs of the *AbGTs* were amplified by PCR using gene-specific primer pairs.

**Expression and purification of AbCGT**. The *AbCGT* gene was amplified using the AbCGT-F and AbCGT-R primer pair (Supplementary Table 5). After verification of the sequence, the coding region of AbCGT was then amplified using the AbCGT-F$_{pET28a}$ and AbCGT-R$_{pET28a}$ primer pair and inserted into pET28a according to the ClonExpress® II One Step Cloning Kit (Vazyme Biotech, China). The recombinant plasmid was then transformed into *Trans*etta (DE3) *E. coli* (TransGen Biotech, China) for heterologous expression. Luria-Bertani (LB) medium (400 mL) containing 50 μg mL$^{-1}$ kanamycin and 34 μg mL$^{-1}$ chloromycetin was inoculated with 1 mL of an overnight culture of recombinant *E. coli* cells. Cells were grown at 37 °C with shaking (200 rpm) until OD$_{600 nm}$ reached 0.4–0.6. The recombinant N-terminal His$_6$-AbCGT expression was achieved by induction with 0.1 mM Isopropyl β-D-thiogalactoside (IPTG) for 12 h at 18 °C with shaking (200 rpm). Cells were harvested by centrifugation at 8000 × g for 10 min at 4 °C. Pellets were resuspended in 40 mL of chilled binding buffer (20 mM phosphate buffer, 0.5 M NaCl, 20 mM imidazole, pH 7.4) containing 1 mM phenylmethylsulfonyl fluoride (PMSF). Cells were disrupted by sonication in an ice bath, and the cell debris was removed by centrifugation at 10,000 × g and 4 °C for 30 min. The soluble fraction was passed through a 0.22-μm syringe filter unit and applied to 1 mL Ni-NTA resin (GE, USA) loaded in a column, which was pre-equilibrated with binding buffer. The resin was subsequently washed with 10 mL binding buffer under a flow rate of 1 mL min$^{-1}$ at 4 °C. After washing with 5 mL washing buffer (20 mM phosphate buffer, 0.5 M NaCl, and 100 mM imidazole; pH 7.4), elution was performed with 5 mL of elution buffer (20 mM phosphate buffer, 0.5 M NaCl, and 200 mM imidazole; pH 7.4). Finally, the recombinant protein was desalted in the desalting buffer (50 mM Tris-HCl buffer, 50 mM NaCl, 1 mM DTT, and 1% glycerol; pH 7.4) by concentration and dilution using an Amicon Ultra-30K (Millipore, USA). Protein purity was confirmed by SDS-PAGE, and protein concentration was determined by the Protein Quantitative Kit (Bradford) (TransGen Biotech, China). The approximate protein yield of AbCGT (50.8 kDa) was 6.9 mg L$^{-1}$.

**Glycosyltransferase activity assays**. The reaction mixture contained 0.5 mM sugar donors (1.0 mM for di-glycosylation), 0.25 mM aglycons, 50 mM Tris-HCl (pH 7.4), and 50–100 μg of purified AbCGT in a final volume of 100 μL. Due to the instability of some acceptors in the buffer of pH 11.0, the reactions were performed under the physiological conditions (pH 7.4, 30 °C) to compare the catalytic activities of AbGT27 with different substrates under the same conditions. The reactions were performed at 30 °C for up to 12 h and terminated by the addition of 200 μL of ice-cold methanol. Subsequently, samples were centrifuged at 15,000 × g for 30 min to collect the supernatant, and aliquots were analyzed by HPLC-UV/ESIMS. HPLC analysis was performed on a Shiseido CAPCELL PAK C18 MG III column (250 mm × 4.6 mm I.D., 5 μm, Shiseido Co., Ltd., Japan) at a flow rate of 1 mL min$^{-1}$. The mobile phase was a gradient elution of solvents A (0.1% formic acid aqueous solution) and B (methanol). The gradient programs were used for the analysis of the reactions (Supplementary Table 6). The conversion rates of the enzyme reactions were calculated from peak areas of glycosylated products and substrates as analyzed by HPLC. To facilitate inferential statistical analysis, three parallel assays were routinely performed; the means ± SD from triplicate analyses are reported here.

**Effects of pH, temperature, and divalent metal ions in enzymatic reactions**. To determine the optimal pH, the enzymatic reaction was performed at 30 °C in various reaction buffers with pH values in the ranges of 5.0−6.0 (citric acid–sodium citrate buffer), 6.0−7.0 (Na$_2$HPO$_4$–NaH$_2$PO$_4$ buffer), 7.0–9.0 (Tris-HCl buffer) and 9.0–12.0 (Na$_2$CO$_3$–NaHCO$_3$ buffer). To assay for the optimal reaction temperature, the reaction mixtures were incubated at pH 11.0 and different temperatures (15−60 °C). To test the necessity of divalent metal ions for AbCGT, BaCl$_2$, CaCl$_2$, CoCl$_2$, CuCl$_2$, FeCl$_2$, MgCl$_2$, MnCl$_2$, ZnCl$_2$, and EDTA were

used individually at a final concentration of 5 mM. The reactions were performed with phloretin (**17**, 0.25 mM) as the acceptor and UDP-Glc (1.0 mM) as the sugar donor and 5 µg of purified AbCGT in a final volume of 100 µL for 10 min. For quantification, three parallel assays were routinely carried out.

**Determination of kinetic parameters of AbCGT**. Assays were performed in a final volume of 100 µL 50 mM Tris-HCl (pH 7.4) at 30 °C, and contained purified recombinant AbCGT (1.5 µg) and saturating UDP-Glc (5 mM) while varying phloracetophene (**11**) and phloretin (**17**) concentrations (5–500 µM), respectively. The reactions terminated by the addition of 200 µL of ice-cold methanol. Subsequently, samples were centrifuged at $15{,}000 \times g$ for 30 min to collect the supernatant, and aliquots were analyzed by HPLC as described above. All experiments were performed in triplicate. The value of $K_m$ was calculated with the method of Lineweaver–Burk plot.

**Preparation of potential acceptors of AbCGT by chemical synthesis**. In brief, a solution of phloroglucinol (2.0 mmol) and $K_2CO_3$ or $Cs_2CO_3$ (1.0 mmol) in DMF (1.2 mL) was stirred vigorously at 0 °C for 10 min. Various bromine substituents (1.0 mmol) in DMF (1.2 mL) were added in a dropwise manner. The reaction mixture was stirred at 0 °C for 12 h and then poured into water (6 mL) and extracted with EtOAc (3 mL × 2). The combined organic phases were washed with brine (3 mL) and water (3 mL) successively. The organic layer was then dried ($Na_2SO_4$) and concentrated, and the residue was purified by semi-HPLC. The detailed experimental process and yields of the prepared aglycons are shown in the Supplementary Information.

**Scale-up enzymatic reactions**. Generally, 10–30 µmol of aglycon was dissolved in 1 mL of DMSO and diluted with buffer solution (50 mM Tris HCl, pH 7.4; total volume 25 mL). UDP-Glc (20–60 µmol) was added along with 20 mg of purified AbCGT. Due to the high cost of UDP-Glc, excess purified AbCGT was used to make sure the full use of UDP-Glc in vitro enzymatic reactions. The reactions were performed at 30 °C for 12 h followed by extraction five times with $5 \times 50$ mL of ethyl acetate. The organic phase was pooled and evaporated to dryness under reduced pressure. For phloracetophene (**11**) and phloretin (**17**), the reaction mixtures were directly evaporated to dryness under reduced pressure. The residue was then dissolved in 1.5 mL of methanol and purified by reverse-phase semi-preparative HPLC. The obtained products were analyzed by MS and NMR.

**Glycosylation assays catalyzed by whole-cell biocatalyst *E. coli-AbCGT***. The induced cells of *E. coli-AbCGT* were harvested by centrifugation at $3000 \times g$ for 5 min at 4 °C and washed with ddH₂O. Pellets were resuspended by M9 minimal medium containing 2% glucose to appropriate cell density ($OD_{600} = 3.0$). Prepared whole-cell biocatalyst of *E. coli-AbCGT* (500 µL) was then added to 2-mL tubes containing various acceptors at a concentration of 0.25 mM. The mixture was incubated at 30 °C for 12 h with shaking (200 rpm). The reactions were analyzed using GT activity assays as described above. For quantification, three parallel assays were routinely carried out.

**Optimization of cell density and substrate concentration in whole-cell biocatalysis**. To study the optimal cell density, the reaction was performed in various cell densities with $OD_{600}$ values in the range of 1.0–15.0 and a substrate concentration of 150 mg L⁻¹. To assay for the optimal substrate concentration, the reactions were performed with 50–500 mg L⁻¹ of aglycon **46** at $OD_{600} = 13.0$. For quantification, three parallel assays were routinely carried out.

**Investigation of the recyclability and stability of the whole-cell biocatalyst**. For the first round of catalytic reaction, 500 mL of engineered cell biocatalyst of *E. coli-AbCGT* ($OD_{600} = 13.0$) was mixed with 150 mg of **46** and incubated at 25 °C for 12 h with shaking (180 rpm). Cells were then collected by centrifugation at $3000 \times g$ for 5 min and washed with M9 minimal medium. Pellets were resuspended with M9 minimal medium and reused in the second round as described in the first round. In total, ten rounds of reactions were performed to evaluate the recyclability and stability of the engineered cell biocatalyst in consecutively catalyzing reactions. The concentrations of **46** and **46a** were obtained according to their corresponding standard curves, and three parallel assays were routinely carried out.

**Inhibitory activity and selectivity assay against SGLT2 for *C*-glycosides in vitro**. Human full-length SGLT2 cDNA (NM_003041.3) was synthesized by Invitrogen Co., Ltd (Shanghai, China) and amplified with a forward primer (5′-CCGCTCGAGGCCACCATGGACAGTAGCACCTGGAGC-3′) and reverse primer (5′-CCGGAATTCTCAGGCAAAATATGCATGGCAAAAG-3′). The PCR products were cloned into the retrovirus vector pMSCVpuro at the *Xho* I and *EcoR* I sites generating pMSCVpuro-SGLT2, and the sequences were confirmed by Sanger sequencing. To produce retrovirus, a mixture containing the plasmid pMSCVpuro-SGLT2 and the packing plasmids (Gag-Pol and pVSV-G) were transfected into 293T cells with Neofect (Neobiotech Co Ltd, Korea). Culture medium was changed

after 16 h post transfection. The supernatant containing retrovirus was collected after 48 h post transfection and filtered through a 0.45-µm PVDF filter membrane. For generating the cells stably expressing SGLT2, HEK293 cells were transduced with retrovirus. After 48 h post infection, puromycin (2 µg mL⁻¹) was used to screen the positive clone cell lines for 2 days, and the SGLT2 expression status was validated by western blot analysis[46]. To evaluate the selectivity of compounds, pMSCVpuro-SGLT1 stably transfected HEK293 cell line was also constructed in the same way. The full-length SGLT1 cDNA was obtained by PCR with forward primer: 5′-CCGCTCGAGGCCACCATGGACAGTAGCACCTGGAGC-3′ and reverse primer: 5′-CCGGAATTCTCAGGCAAAATATGCATGGCAAAAG-3′.

D-glucose transport was measured by performing 100 µM 1-[N-(7-Nitrobenz-2-oxa-1,3-diazol-4-yl)amino]-1-deoxy-D-glucose (1-NBDG) uptakes in cells overexpressing SGLT1 or SGLT2. Cells were seeded in 24-well plates (10,0000 cells/well), grown to 90–95% confluency, and then the transport assays were performed. Cells were incubated in transport $Na^+$ buffer containing 120 mM NaCl, 4.7 mM KCl, 1.2 mM MgCl₂, 2.2 mM CaCl₂, 10 mM HEPES, pH 7.4 with various concentrations of *C*-glycosides. For nonspecific transport determination, cells were incubated in $Na^+$-free buffer containing 140 mM choline chloride, 4.7 mM KCl, 1.2 mM MgCl₂, 2.2 mM CaCl₂, 10 mM HEPES, pH 7.4. After 4 h-incubation at 37 °C, cells were washed twice with ice-cold stop buffer ($Na_+$-free buffer containing 0.5 mM phlorizin), incubated at room temperature in cell lysis buffer (0.1 mM NaOH), and then neutralized with 0.1 mM HCl. Fluorescence intensity at 485/535 nm was then measured to determine the inhibitory activity of SGLT2 for *C*-glycosides. All experiments were repeated in triplicate. The $IC_{50}$ values were determined using nonlinear regression with Prism 5 software.

**Evaluation of the hypoglycemic effect of 46a in vivo**. The hypoglycemic effect of **46a** was firstly evaluated by oral glucose-tolerance test (OGTT) in mice with normal blood glucose levels. Male Institute of Cancer Research (ICR) mice weighing 24–26 g were obtained from Beijing Vital River Laboratory Animal Technology Co., Ltd (Beijing, China) and divided into three groups ($n = 7$–8). Briefly, ICR mice were orally administered vehicle or test compounds (5 mg kg⁻¹ dapagliflozin and 100 mg kg⁻¹ **46a**). After 2 h of fasting, blood samples were collected from the tails for determination of baseline values of blood glucose (0 min). The mice were then orally administered glucose (2 g kg⁻¹ BW), and blood samples were collected at 15, 30, 60, and 120 min. A glucose oxidase (GOD) assay was used for blood glucose-level measurement. The values of area under the glucose-time curve (AUC) were calculated.

Type 1 diabetes mellitus (T1DM) mice were established by intravenous injection of alloxan tetrahydrate (68 mg kg⁻¹ BW). Mice with serum glucose >200 mg dL⁻¹ 72 h after alloxan injection were considered diabetic and randomly divided into three groups ($n = 8$) as follows: the alloxan-induced DM group, positive drug group (dapagliflozin, 5 mg kg⁻¹ BW), and **46a** treatment group (100 mg kg⁻¹ BW). Nonfasting blood glucose was kinetically monitored at 1, 2, 4, 6, 8, 24, and 48 h after treatment with **46a**. AUC values during 0–8 h, 8–24 h, and 24–48 h were calculated.

To evaluate the hypoglycemic effect of **46a** in type 2 diabetes mellitus (T2DM), female KKAy mice weighing 42–48 g were obtained from the Animal Center of the Institute of Laboratory Animal Sciences, Chinese Academy of Medical Sciences and Peking Union Medical College. The model mice were divided into the following three groups ($n = 6$–7): model control KKAy, dapagliflozin (5 mg kg⁻¹ BW), and **46a** (100 mg kg⁻¹ BW). Blood samples were collected at 1, 2, 4, 6, 8, 24, and 48 h after treatment. AUC values during 0–8 h, 8–24 h, and 24–48 h were calculated. The glucose levels in urine 4 h and 6 h after administration were also determined using a Biosen C-Line Glucose analyzer (EKF Diagnostics, Germany).

All mice were housed in a temperature- and humidity-controlled room and allowed free access to standard chow diet for 1 week before the study. All experiments were complied with all relevant ethical regulations and approved by the Institutional Animal Care and Use Committee of the Institute of Materia Medica, Chinese Academy of Medical Sciences.

**Homology modeling and molecular docking**. To build the 3D structure of AbCGT binding with 2-hydroxynaringenin (**23**), homology modeling was performed using SWISS-MODEL (https://swissmodel.expasy.org/) based on the GgCGT crystal structure (PDB code 6L5R.1)[32]. Molecular docking between AbCGT and 2-hydroxynaringenin was investigated using AutoDockTools-1.5.6. All docking calculations were restricted to the predicted binding pocket by defining the active site with residue His19, F89, F90, and F194.

**Site-saturation mutagenesis of AbCGT**. Site-saturation mutagenesis of AbCGT at residue 183 was performed by PCR using pET28a-*AbCGT* as a template and the corresponding degenerate primers, which are listed in Supplementary Table 5. PCR products were purified and transformed into *Trans*1-T1 (DE3) *E. coli*. The sequences of pET28a-*AbCGT* mutants were confirmed by sequencing, and transformed into *Trans*etta (DE3) *E. coli* for heterologous expression following protein purification.

**Reporting summary**. Further information on research design is available in the Nature Research Reporting Summary linked to this article.

## Data availability

The authors declare that all data supporting the findings of this study are available within the paper and its Supplementary Information Files, including experimental details, characterization data, and $^1$H and $^{13}$C NMR spectra (Supplementary Figs. 59–166) of all prepared compounds. All the data are available from the authors upon reasonable request. The gene sequence of the AbCGT is deposited in GenBank, accession no. MN747045. Source data are provided with this paper.

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

## Acknowledgements

We acknowledge the National Natural Science Foundation of China (81602999 to K.X.), the Drug Innovation Major Project (2018ZX09711001-001-006 to J.D. and 2018ZX09711001-003-005 to F.Y.), the CAMS Innovation Fund for Medical Sciences (CIFMS-2016-I2M-3-012 to J.D. and F.Y.), PUMC Disciplinary Development of Synthetic Biology (201920100801 to J.D.), and the National Infrastructure of Microbial Resources (NIMR-2017-3 to J.D.).

## Author contributions

K.X., J.D., and F.Y. conceived this project. K.X. conducted the cloning and expression of the GTs gene, protein purification, biochemical kinetics, chemo-biocatalysis, and isolation and structural determination of substrates and products. S.S. contributed to isolate the enzymatic products. X.Z. and F.Y. contributed to bioassay of the SGLT2 inhibitory activity of *C*-glycosides in vitro and in vivo. K.X. performed the homology modeling, molecular docking, and saturation mutation. All authors discussed the results and wrote the paper.

## Competing interests

The authors declare no competing interests.
