## [Peer Review File · Nature Communications]

REVIEWER COMMENTS

Reviewer #1 (Remarks to the Author):

Dear Editor,

I am satisfied with the feedback given by the authors and in my opinion the manuscript is ready for publication.

Reviewer #2 (Remarks to the Author):

This work presents a comprehensive research. After revision, the manuscript was improved and all the comments from the reviewers were addressed.

Reviewer #3 (Remarks to the Author):

In their resubmission, the authors have addressed certain concerns by all four referees, and have certainly made an effort to improve the manuscript. However, certain issues persist, the biggest one being that crucial information is still left out of the manuscript. For instance:

1) The authors have re-named compounds with mono- and di-glycosylation in Fig. 1, but they still have not depicted the yields of these compounds here.

2) The authors have included MS2 fragmentation patterns, but it is not entirely clear why these fragmentation patterns are different (i.e. what are M-90 and M-120?), which is important because the authors seem to rely a lot on these. This should be confirmed with reference compounds.

3) The text indicates that MS data for compounds 20a and 21a exist, but these are not in the SI. Similarly, there is no MS data of compound 11.

4) In Figure S27, compound 26 gave multiple peaks in the MS/MS data. Among them is the [M+H-120]⁺ peak (m/z = 218) which according to the authors should be the C-glycoside. But in figure 1 they claim this compound is only converted to the N-glycoside.

5) Table R2 helps to distinguish between the glycosylation products using UDP-Glc vs UDP-Gal, but the authors do not overlay these with the compound isolated from turnover in *E. coli*. This is important.

6) All tables and figures provided to the referees should be included in the paper. There is a lot of ambiguity that is partially addressed by these (and Table R2 has to be extended, as stated above).

7) In their response, the authors state that they used an excess of enzyme due to the cost of UDP-Glc. This has to be mentioned in the manuscript, as it leaves questions about enzyme quantities otherwise.

8) Concerning the acceptor library that was used, the authors should state explicitly which ones were commercial vs. self-made.

9) The methods added on cell culture is still not sufficient to reproduce the experiments. For instance:

- "The target gene was connected to the vector " - how? cloning strategy, enzymes used for ligation etc.?

- "Then the sequenced plasmid was transfected into HEK293 cells , and puromycin was used to screen the positive clone cell lines. Western blot and immunofluorescence analysis were performed to evaluate the expression of SGLT2 in stable transfected cells." - how was transfection done? How much puromycin, selection for how long? How was Western Blot done, which antibodies, where is the data?

- "D-glucose transport was measured by performing 100 μ M 1-[N-(7-Nitrobenz-2-oxa-1,3-diazol-4-yl)amino]-1-deoxy-D-glucose (1-NBDG) uptakes in cells over-expressing SGLT1 or SGLT2 cultured on 24-well plates. Cells were incubated..." how many cells? Confluency?

- "To evaluate the selectivity of compounds, pMSCVpuro-SGLT1 stably transfected HEK293 cell line was also constructed" - the same way? Experimental details?

- "293T-Gluc cells were generated by transfection of plasmid DNA pLenti6-Gluc constitutively..." - how?

- "Cell viability was evaluated by cell counting kit-8 (CCK-8) assay. Briefly, 293T cells were cultured in a 96-well plate and incubated with compounds." - how many cells? Confluency?

10) The authors use the term "catalytic selectivity" in two very different contexts, either for the construction of O- vs. C- glycosides and to distinguish between acceptors. They should not use this term interchangeably.

11) Finally, the authors have made use of the promiscuous selectivity of GTs to make potential new drugs. There are certain aspects here that are of interest and novel. But both the experimental approach (cf. the data missing on turnover to mono- vs. diglycosylated products) and the application seem to be limited. To the latter point, the authors have shown application of their compounds, but have not mentioned what the difference (or, potentially, the advantage) would be to marketed drugs such as the FDA-approved Empagliflozin. Furthermore, they have not addressed that there are other SGLTs - would there be side reactions pertaining to inhibition of these?

Taken together, there are quite a few flaws in the manuscript that may not be solved by another round of revision, and may be addressed better by potentially publishing two separate papers on the synthetic and application aspects.

Reviewer #4 (Remarks to the Author):

In this manuscript, the authors presented AbcGT is promiscuous GT, has the ability for C-, N-, S-, and O- glycosylation, accepts phenolic substrates without acyl group, and can use diverse sugar donors. The authors used E. coli as a whole-cell system for the chemoenzymatic synthesis of C-glycosides and some new glycosides demonstrate SGLT2 inhibition. It is an impressive finding however, there are some inconsistencies and phrasing that can be misleading.

- The authors refer to ref:44 for homology modeling in the "Methods" section and Figure 6A. However, the text describes the homology model was based on UGT85H2 (lines: 326-328).
- Based on the ref:44, GgCGT is highly promiscuous C-GT, has the ability to C-, N-, S-, and O-glycosylation. While the introduction and conclusion section mention MiCGT and TcCGT1, no discussion or comparison with GgCGT has been made throughout the text.
- While Figure 1A shows several sugar donors were tested with compound 17 and three sugar donors resulted in 100% conversion (one was almost 90%), the text does not give much description about the reason for using several sugar donors nor does it discuss the importance of

the results, or their utility in synthesizing sugar derivatives of SGLT2 inhibitors. The sugar-donors study stands by itself without having much relevance to the rest of the text.

- The "chemoenzymatic" word is used both as hyphenated and non-hyphenated- please choose one.
- Line 29: "the key sites" - this should be key residues.
- Lines 93-97 - this MS/MS fragmentation pattern requires a reference.
- It appears that Mn²⁺ and Mg²⁺ increase the yield by 50%- yet the enzyme reaction conditions mentioned did not use this any divalent metal ion- an explanation for not using divalent metal ion would be helpful.
- Lines 138-139: " albeit not for the native substrates" - in the absence of known substrates for the enzyme, this does not look appropriate.
- Line 164: "high efficiency" - In enzyme biochemistry "efficiency" refers to K_{cat}/K_M and not %conversion in 12h assay.
- Line 171: "C-glycosylation site needs at least two hydroxyl groups in its ortho positions" - this is misleading as it sounds as if hydroxyl groups are needed on both sides of C-glycosylation site.
- Lines 216-222: The description of the first mention of a compound should follow the ascending order of the compound's number.
- Line 224: organize the compounds in ascending order of compound numbers when they are referred together.
- Line 250- order the compounds within the parenthesis as per the compound number
- Line 325: "key amino acids"- change to "key amino acid residues"
- Lines 330, 353: "key amino acid" -change to "key amino acid residue"
- Supporting Information- HPLC-Mass fragmentation pattern needed to identify 1a, 2a and 3a is almost invisible- please provide better data
- Supporting Information -HPLC-MS/MS fragmentation chromatograms are missing for compounds 9, 16, 19, 20, 21, 28, 30, 38, 39, 42, 43 and 44. These are needed even though NMR information has been provided.
- Supporting Information - Page S24-S35- please order the data as per compound numbers
- Supporting Information - Page S46-S77- please order the data as per compound numbers
- Supporting Information - Pages S123 and S126- please order the data as per compound numbers

Responses to Reviewers #1

Reviewer #1 (Remarks to the Author):

I am satisfied with the feedback given by the authors and in my opinion the manuscript is ready for publication.

We would like to thank this reviewer for his/her time to review our manuscript. His/her previous suggestions and comments are very helpful to improve the quality of our manuscript.

Responses to Reviewers #2

Reviewer #2 (Remarks to the Author):

This work presents a comprehensive research. After revision, the manuscript was improved and all the comments from the reviewers were addressed.

We are appreciated for the positive comments and helpful suggestions by this reviewer, which improved the overall quality of our work greatly.

Responses to Reviewers #3

Reviewer #3 (Remarks to the Author):

In their resubmission, the authors have addressed certain concerns by all four referees, and have certainly made an effort to improve the manuscript. However, certain issues persist, the biggest one being that crucial information is still left out of the manuscript.

Thank you very much for your time to re-review our manuscript and give us helpful comments and suggestions. Following these, we have revised accordingly, and our responses are listed below.

For instance:

1) The authors have re-named compounds with mono- and di-glycosylation in Fig. 1, but they still have not depicted the yields of these compounds here.

Response #3-1: Thanks a lot for pointing out this point. We apologize for not providing the yields of mono-*C*- and di-*C*-glycosides in **Figure 1**. So to make this point clear, we have added the yields of both mono-*C*- and di-*C*-glycosides in **Figure 1**. There are six substrates (**10**, **11** and **16–19**) that can be di-*C*-glycosylated (light blue) in **Figure 1**. Besides, their corresponding HPLC chromatograms with the yields of mono-*C*- and di-*C*-glycosides were also shown in the following figures (**Figures r1–r6**), which have been added in the **Supplementary Information (Figures S2, S3, S23 and S27–S29, SI)**. Thanks so much for the suggestions.

Figure 1. Probing the aglycon acceptor and sugar donor promiscuity of AbCGT.

A) Percent conversion of each aglycon acceptor with AbCGT. The aglycon acceptors and sugar donors are listed based on the structural scaffolds shown in part B and C. The red, blue, light blue, green and black columns represent conversion rates of *O*-, mono-*C*-, di-*C*-, *S*- and *N*-glycosylation reactions, respectively. B) The structures of the aglycon acceptors and corresponding glycosylated products prepared from the scale-up enzymatic reactions. C) The structures of sugar donors used in this work. The sugar donor promiscuity was tested with phloretin (**17**) as the acceptor. ND: not detected. Experiments were performed at pH 7.4 and 30 °C for up to 12 h in triplicate, and the standard deviation is noted.

Figure r1. HPLC-UV/ESI-MS analysis of AbCGT enzyme product using aglycon **10** and UDPG as substrates. A) HPLC-UV analysis of the AbCGT catalyzing reaction; B) Typical negative ion MS for the peak of **10aa**; C) Typical negative MS² for peak of **10aa**. (Figure S23 in the Supplementary Information)

Figure r2. HPLC-UV/ESI-MS analysis of AbCGT enzyme product using phloracetophene (**11**) and UDPG as substrates. A) HPLC-UV analysis of the AbCGT catalyzing reaction; B) Typical negative ion MS for the peak of **11aa**; C) Typical negative MS² for peak of **11aa**. (Figure S2, SI)

Figure r3. HPLC-UV/ESI-MS analysis of AbCGT enzyme product using aglycon **16** and UDPG as substrates. A) HPLC-UV analysis of the AbCGT catalyzing reaction; B) Typical negative ion MS for the peak of **16aa**; C) Typical negative ion MS² for the peak of **16aa**. (Figure S27, SI)

Figure r4. HPLC-UV/ESI-MS analysis of AbCGT enzyme product using phloretin (**17**) and UDPG as substrates. A) HPLC-UV analysis of the AbCGT catalyzing reaction; B) Typical negative ion MS for the peak of **17aa**; C) Typical negative MS² for peak of **17aa**. (Figure S3, SI)

Figure r5. HPLC-UV/ESI-MS analysis of AbCGT enzyme products using aglycon **18** and UDPG as substrates. A) HPLC-UV analysis of the AbCGT catalyzing reaction. The total conversion rate and the yields of mono-C-glycoside (**18a**) and di-C-glycoside (**18aa**) are shown, respectively; B) Typical negative ion MS for the peak of **18aa**; C) Typical negative MS² for peak of **18aa**; D) Typical negative ion MS for the peak of **18a**; E) Typical negative MS² for peak of **18a**. (Figure S28, SI)

Figure r6. HPLC-UV/ESI-MS analysis of AbCGT enzyme products using aglycon **19** and UDPG as substrates. A) HPLC-UV analysis of the AbCGT catalyzing reaction. The total conversion rate and the yields of mono-C-glycoside (**19a**) and di-C-glycoside (**19aa**) are shown; B) Typical negative ion MS for the peak of **19a**; C) Typical negative MS² for peak of **19a**; D) Typical negative ion MS for the peak of **19aa**; E) Typical negative MS² for peak of **19aa**. (Figure S29, SI)

2) The authors have included MS² fragmentation patterns, but it is not entirely clear why these fragmentation patterns are different (i.e. what are M-90 and M-120?), which is important because the authors seem to rely a lot on these. This should be confirmed with reference compounds.

Response #3-2: Thanks for the expertise of this reviewer. Generally, the C-glucosides and O-glucosides exhibited different MS² fragmentation patterns due to their different glycosidic bonds. The examples were shown in **Figure r7**.

Figure r7. MS/MS fragmentation of C-glucoside (**1**) and O-glucoside (**2**). A) The characteristic MS/MS fragment ions of C-glucoside (**1**); B) Typical negative ion MS for C-glucoside (**1**); C) Typical negative MS/MS for C-glucoside (**1**); D) The characteristic MS/MS fragment ions of O-glucoside (**2**); E) Typical negative ion MS for O-glucoside (**2**); F) Typical negative MS/MS for O-glucoside (**2**).

For the glycosidic bonds, C-C bonds are more stable than C-O bonds in the MS/MS analysis. As shown in **Figure r7**, the neutral loss of 120 and 90 Da with high intensity represented the present of C-glycosidic bonds (**Figure r7A**) and the high intensity of

neutral loss of 162 Da suggested the existence of *O*-glycosidic bonds (**Figure r7D**). Following the reviewer's suggestions, this information has been also confirmed by using *C*-glucoside **1** and *O*-glucoside **2** as the reference compounds (**Figure r7B, r7C, r7E and r7F**).

In the case of *O*-glucoside **2** in **Fig r7**, the ionization energy caused the ether glycosidic bond to fragment, with the associated loss of 162 Da to give m/z 273. In contrast, the C-C-linked glucoside **1** gave a very different fragmentation, with characteristic losses of 90 and 120 Da as opposed to the cleavage of the intact pyranoid ring (loss of 162 Da) (ref1–3).

Therefore, the characteristic fragmentation of the *C*-glucoside is $[M-H-90]^-$ and $[M-H-120]^-$ while that of the *O*-glucoside is $[M-H-162]^-$. We have added the related references in the manuscript. The related references (ref1–3, Line 98) have been added in the manuscript.

[ref1] Brazier-Hicks, M. *et al.* The *C*-glycosylation of flavonoids in cereals. *J. Biol. Chem.*, **284**, 17926–17934 (2009).

[ref2] Kazuno, S., Yanagida, M., Shindo, N. & Murayama, K. Mass spectrometric identification and quantification of glycosyl flavonoids, including dihydrochalcones with neutral loss scan mode. *Anal. Biochem.*, **347**, 182–192 (2005).

[ref3] Ferreres, F., Silva, B. M., Andrade, P. B., Seabra, R. M. & Ferreira, M. A. Approach to the study of *C*-glycosyl flavones by ion trap HPLC-PAD-ESI/MS/MS: application to seeds of quince (*Cydonia oblonga*). *Phytochem. Anal.*, **14**, 352–359 (2003).

3) The text indicates that MS data for compounds **20a** and **21a** exist, but these are not in the SI. Similarly, there is no MS data of compound **11**.

Response #3-3: Thanks for pointing out this point for us. We have provided the HPLC-MS data for the AbCGT catalyzing reactions with substrates **11** (**Figure r2**, in Responses to **Reviewers #3-1**), **20** and **21** (**Figures r19** and **r20**, in Responses to **Reviewers #4-16**), whose glycosylated products have been prepared from the scale-up reactions and characterized by NMR in the SI. This information was also

added to the supplementary information (**Figures S2, S30 and S31, SI**).

4) In **Figure S27**, compound **26** gave multiple peaks in the MS/MS data. Among them is the $[M+H-120]^+$ peak ($m/z = 218$) which according to the authors should be the C-glycoside. But in **Figure 1** they claim this compound is only converted to the N-glycoside.

Response #3-4: Thanks for such good question. As we described in **Reviewer #3-2**, the characteristic fragmentation of the C-glycoside is $[M-H-90]^-$ and $[M-H-120]^-$ while that of the O-glycoside is $[M-H-162]^-$. Generally, the characteristic fragmentations of C-glycosides and other types of glucosides (O-, S-, N-glucosides) represent fragmentations with high intensity in MS/MS. The fragmentation $[M-H-120]^-$ might be found in MS/MS of O-, S- and N-glucosides. The fragmentation $[M-H-162]^-$ might also be found in MS/MS of C-glycosides. However, these are not the major fragmentations and only observed in a low intensity. In the positive mode MS/MS for **26a**, although the peak of $[M+H-120]^+$ was observed, the major fragmentation was $[M+H-162]^+$. The fragmentation of **26a** was coincidence with an N-glycoside.

To further confirm the structure, **26a** was compared with the standard 4-Methylcoumarin-7-amino- β -D-glucoside, which was prepared and characterized by NMR in our previous work (Xie K, et al. Enzymatic N-Glycosylation of Diverse Arylamine Aglycones by a Promiscuous Glycosyltransferase from *Carthamus tinctorius*. *Advanced Synthesis & Catalysis*, **2017**, 359, 603). The HPLC-MS/MS analysis showed that **26a** and 4-methylcoumarin-7-amino- β -D-glucoside exhibited the same retention time and MS/MS fragmentation (**Figure r8**). This information was also shown in **Figure S36** in the SI.

Figure r8. HPLC-UV/ESI-MS analysis of AbCGT enzyme product using aglycon **26** and UDPG as substrates. A) HPLC-UV analysis of the AbCGT catalyzing reaction; B) HPLC-UV analysis of the standard 4-Methylcoumarin-7-amino- β -D-glucoside; C) Typical positive ion MS for the peak of **26a**; D) Typical positive MS^2 for peak of **26a**. E) Typical positive ion MS for the peak of 4-Methylcoumarin-7-amino- β -D-glucoside; F) Typical positive MS^2 for peak of 4-Methylcoumarin-7-amino- β -D-glucoside. (Figure S36, SI)

5) Table R2 helps to distinguish between the glycosylation products using UDP-Glc vs UDP-Gal, but the authors do not overlay these with the compound isolated from turnover in *E. coli*. This is important.

Response #3-5: Thanks for the helpful advice. Yes, **Table R2** and **Figure R7 (Figure r9** in the new version of Responses) can help to distinguish between the glycosylation products using UDP-Glc and other UDP-sugars. According to the suggestion, we have added the HPLC chromatogram of reactions catalyzed by whole cell biocatalyst. These results showed that the glycosylated products of enzyme catalyzing reactions with UDP-Glc as the donor and whole cell catalyzing reactions are both C-glycosylated. The **Table** and **Figures** have been added to the supplementary information (Table S3 and Figure S9, SI).

Figure r9. HPLC analysis of AbCGT catalyzing reactions *in vitro* and *in vivo*. A) AbCGT catalyzing reactions *in vitro* with UDP-Glc as the sugar donor; B) AbCGT catalyzing reactions *in vitro* with UDP-Gal as the sugar donor; C) whole cell catalyzing reactions *in vivo*. Phloretin (**17**) was used as an acceptor. (**Figure S9, SI**)

6) All tables and figures provided to the referees should be included in the paper. There is a lot of ambiguity that is partially addressed by these (and Table R2 has to be extended, as stated above).

Response #3-6: Thanks for the helpful suggestion. The tables and figures including **Table R2** provided to the referees have been added to the paper as described in the

responses. (Table S3, Figures S9, S14–S16, S22, S27–S31, S36, S38, S40, S48, S49, S52–S54 and S164–S169 were provided in the SI and the corresponding information and explanation have been also added in the manuscript.)

7) In their response, the authors state that they used an excess of enzyme due to the cost of UDP-Glc. This has to be mentioned in the manuscript, as it leaves questions about enzyme quantities otherwise.

Response #3-7: Thanks for the helpful advice. The using of an excess of enzyme due to the cost of UDP-Glc has been be mentioned in the manuscript.

Lines 520–521 (in the new version): *“Due to the high cost of UDP-Glc, excess purified AbCGT was used to make sure the full use of UDP-Glc in vitro enzymatic reactions.”*

8) Concerning the acceptor library that was used, the authors should state explicitly which ones were commercial vs. self-made.

Response #3-8: Following the suggestion, we have state explicitly about the members of the acceptor library in the manuscript.

Lines 416–422 (in the new version): *“Chemicals and reagents including commercially available aglycons (1–15, 17, 19, and 23–29) and sugar donors were purchased from Sigma-Aldrich (St. Louis, USA), J&K Scientific Ltd (Beijing, China), InnoChem Science & Technology Co., Ltd (Beijing, China), BioBioPha (Kunming, China), Beijing BG Biotech Co., Ltd (Beijing, China) and Beijing Adhoc International Technologies Co., Ltd (Beijing, China). Aglycons 16, 18 and 30 were chemically synthesized as described previously¹⁵ and aglycons 20–22 and 31–54 were synthesized in this work.”*

9) The methods added on cell culture is still not sufficient to reproduce the experiments. For instance:

Response #3-9: Thanks for the helpful comments and questions. To make the methods about the cell culture clear and sufficient enough to reproduce the experiments we have rewritten this part (Lines 553–588, in the new version). The detailed experimental procedure was added to the manuscript.

Lines 553–588 (in the present version):

“Inhibitory activity and selectivity assay against SGLT2 for C-glycosides in vitro

Human full length SGLT2 cDNA (NM_003041.3) was synthesized by Invitrogen Co., Ltd (Shanghai, China) and amplified with forward primer (5'-CCGCTCGAGGCCACCATGGACAGTAGCACCTGGAGC-3') and reverse primer (5'-CCGGAATTCTCAGGCAAAATATGCATGGCAAAAG-3'). The PCR products were cloned into the retrovirus vector pMSCVpuro at the Xho I and EcoR I sites generating pMSCVpuro-SGLT2 and the sequences were confirmed by Sanger sequencing. To produce retrovirus, a mixture containing the plasmid pMSCVpuro-SGLT2 and the packing plasmids (Gag-Pol and pVSV-G) were transfected into 293T cells with Neofect (Neobiotech Co Ltd, Korea). Culture medium was changed after 16 h post transfection. The supernatant containing retrovirus was collected after 48 h post transfection and filtered through a 0.45 μm PVDF filter membrane. For generating the cells stably expressing SGLT2, HEK293 cells were transduced with retrovirus. After 48 h post infection, puromycin ($2 \mu\text{g mL}^{-1}$) was used to screen the positive clone cell lines for 2 days and the SGLT2 expression status was validated by Western blot analysis.⁴⁶ To evaluate the selectivity of compounds, pMSCVpuro-SGLT1 stably transfected HEK293 cell line was also constructed in the same way. The full length SGLT1 cDNA was obtained by PCR with forward primer: 5'-CCGCTCGAGGCCACCATGGACAGTAGCACCTGGAGC-3' and reverse primer: 5'-CCGGAATTCTCAGGCAAAATATGCATGGCAAAAG-3').

D-glucose transport was measured by performing 100 μM 1-[N-(7-Nitrobenz-2-oxa-1,3-diazol-4-yl)amino]-1-deoxy-D-glucose (1-NBDG) uptakes in cells over-expressing SGLT1 or SGLT2. Cells were seeded in 24-well plates (10,000 cells/well), grown to 90~95% confluency, and then the transport assays were performed. Cells were incubated in transport Na^+ buffer containing 120

mM NaCl, 4.7 mM KCl, 1.2 mM MgCl₂, 2.2 mM CaCl₂, 10 mM Hepes, pH 7.4 with various concentrations of C-glycosides. For nonspecific transport determination, cells were incubated in Na⁺-free buffer containing 140 mM Choline chloride, 4.7 mM KCl, 1.2 mM MgCl₂, 2.2 mM CaCl₂, 10 mM Hepes, pH 7.4. After 4 hour-incubation at 37 °C, cells were washed twice with ice-cold stop buffer (Na₊-free buffer containing 0.5 mM phlorizin), incubated at room temperature in cell lysis buffer (0.1 mM NaOH), and then neutralized with 0.1 mM HCl. Fluorescence intensity at 485/535 nm was then measured to determine the inhibitory activity of SGLT2 for C-glycosides. All experiments were repeated in triplicate. The IC₅₀ values were determined using nonlinear regression with Prism 5 software.”

In addition, some experimental data, which proved the reliability and repeatability of our experiments, were also provided in the following pages.

(1)- "The target gene was connected to the vector " - how? cloning strategy, enzymes used for ligation etc.?"

Response #3-9-1: The target gene and the vector were digested with restriction enzymes *Xho* I and *EcoR* I, respectively, ligated with T4 DNA ligase, and then transformed into Trans1-T1 competent cells (TransGen Biotech, China). Positive clones were confirmed by PCR and sequenced. The detailed information has also been added to the manuscript (Lines 553–588, in the present version).

(2)- "Then the sequenced plasmid was transfected into HEK293 cells , and puromycin was used to screen the positive clone cell lines. Western blot and immunofluorescence analysis were performed to evaluate the expression of SGLT2 in stable transfected cells." - how was transfection done? How much puromycin, selection for how long? How was Western Blot done, which antibodies, where is the data?

Response #3-9-2: The detailed experimental steps and test results are described as follows:

Retrovirus was produced in 293T cells. To produce retrovirus, a mixture containing the plasmid pMSCVpuro-SGLT2 and the packing plasmids (Gag-Pol and pVSV-G)

were transfected into 293T cells with Neofect (Neobiotech Co Ltd, Korea). Culture medium was changed after 16 h post transfection. The supernatant containing retrovirus was collected after 48 h post transfection and filtered through a 0.45 μm PVDF filter membrane. For generating the cells stably expressing SGLT2, HEK293 cells were transduced with retrovirus. After 48 h post infection, puromycin ($2 \mu\text{g mL}^{-1}$) was used to screen the positive clone cell lines for 2 days and the SGLT2 expression status was validated by Western blot analysis (**Figure r10**). The results of western blot and immunofluorescence analysis were previously published (Zhang X L, Wang Y N and Ye F. Development of a method to study the activity and selectivity of SGLT2 inhibitors. *Acta Pharm Sin*, **2017**, 52(6): 897–903.). Anti-SGLT2 antibody was purchased from Santa Cruz Biotechnology, Santa Cruz, CA (sc-47404). As shown in **Figure r10**, SGLT2 protein was over expressed in pMSCVpuro-SGLT2 transfected HEK293 cells and located in both cytoplasm and membrane. The detailed information about the experimental procedure was added to the manuscript (Lines 553–588, in the present version).

Figure r10 Expression of sodium-glucose cotransporter 2 (SGLT2) in stably transfected HEK293 cells. A) SGLT2 expression plasmid digested by restriction enzyme *EcoR* I and *Xho* I, and fragments produced in 0.8% agarose gel electrophoresis. Lane 1: pMSCVpuro; Lane 2: pMSCVpuro-SGLT2; Lane 3: pMSCVpuro digested by *EcoR* I and *Xho* I; Lane 4: pMSCVpuro-SGLT2 digested by *EcoR* I and *Xho* I. M: DNA marker. B) Western blotting analysis for human SGLT2 in the HEK293 cells stable transfected with pMSCVpuro-SGLT2 compared to pMSCVpuro-null transfected cells. Band represents ~70 kDa. C) Immunofluorescence staining of HEK293 cells stably transfected with pMSCVpuro-null (upper panels) and pMSCVpuro-SGLT2 (lower panels) using SGLT2 antibody. Cells were grown on poly-*L*-lysine-coated coverslips for 24 h followed by fixation and immunocytochemistry analysis using CLSM. Green: FITC-tagged hSGLT2; Red: PI counterstain of the nucleus. (Magnification, 1000×)

(3)- ³H-D-glucose transport was measured by performing 100 μM

1-[N-(7-Nitrobenz-2-oxa-1,3-diazol-4-yl)amino]-1-deoxy-D-glucose (1-NBDG) uptakes in cells over-expressing SGLT1 or SGLT2 cultured on 24-well plates. Cells were incubated..." how many cells? Confluency?

Response #3-9-3: Cells were seeded in 24-well plates (10,0000 cells/well), grown to 90~95% confluency. This information has been provided in the manuscript (Lines 577, in the present version).

(4)- "To evaluate the selectivity of compounds, pMSCVpuro-SGLT1 stably transfected HEK293 cell line was also constructed" - the same way? Experimental details?

Response #3-9-4: Thanks for the question. As described in **Response #3-9**, we have provided the experimental details of the construction of pMSCVpuro-SGLT1 stably transfected HEK293 cell line. Yes, we used the same method except the different cell lines (SGLT1 instead of SGLT2). The pMSCVpuro-SGLT1 stably transfected HEK293 cell line was also constructed in the same way as pMSCVpuro-SGLT2 stably transfected HEK293 cell line. As shown in **Figure r11**, SGLT1 was overexpressed in pMSCVpuro-SGLT1 transfected HEK293 cells. These results have also been published in our previous work (Zhang X L, Wang Y N and Ye F. Development of a method to study the activity and selectivity of SGLT2 inhibitors. *Acta Pharm Sin*, **2017**, 52(6): 897–903.). The experimental details have also been described in the manuscript (Lines 553–588 in the present version).

Figure r11 Expression of SGLT1 in stably transfected HEK293 cells. A) RT-PCR analysis of HEK293 cells stably transfected with pMSCVpuro-SGLT1; B) Western blot analysis for human SGLT1. Band represents ~70kDa.

(5)- "293T-Gluc cells were generated by transfection of plasmid DNA pLenti6-Gluc constitutively..." - how?

Response #3-9-5: Thanks for the helpful question. 293T cells were transfected with pLenti6-Gluc using lipofecta-mine2000 in accordance with the manufacturer's protocol, and then were selected with 10 µg/mL Blasticidin 24 h post-transfection. Under antibiotic selective pressure several clonal colonies were obtained and tested for luciferase expression. One clonal cell line demonstrated high-level expression of luciferase; this cell line was named 293T-Gluc and was used for subsequent experiments. For more detailed instructions, please see the reference 1 in the method of “**Experimental Procedures of Anti-influenza A Virus Activity Assays**^[1, 2]” (Page S38, SI). This information has also been added to the method.

Page S37: *“The (Gluc) gene was amplified with forward primer (5′-TATGAATTCGGAAAAACGCCAGCAAC-3′) and reverse primer (5′-ATAAGGGCCCAAATCTTCTTTCATCCGC-3′). PCR products were cloned into pLenti6/V5-DEST vector (Invitrogen) generating pLenti6-Gluc. 293T cells were transfected with pLenti6-Gluc using lipofecta-mine2000 in accordance with the manufacturer's protocol, and then were selected with 10 µg/mL Blasticidin 24 h post-transfection. Under antibiotic selective pressure, several clonal colonies were obtained and tested for luciferase expression. One clonal cell line demonstrated high-level expression of luciferase; this cell line was named 293T-Gluc and was used for subsequent experiments.”*

(6)- "Cell viability was evaluated by cell counting kit-8 (CCK-8) assay. Briefly, 293T cells were cultured in a 96-well plate and incubated with compounds." - how many cells? Confluency?

Response #3-9-6: Thanks a lot for the question. 293T cells were cultured to 80% confluence in a 96-well plate with 1×10^4 cells/well. As suggested, we have added the detailed information to the method. The references (ref1 and 2, SI) were also added in this part.

Experimental Procedures of Anti-influenza A Virus Activity Assays^[1, 2]

...Cell viability was evaluated by Cell Counting Kit-8 (CCK-8) assay. Briefly, 293T cells were cultured to 80% confluence in a 96-well plate with 1×10^4 cells/well and incubated with compounds....

1. Gao Q, Wang Z, Liu Z, et al. A cell-based high-throughput approach to identify

inhibitors of influenza A virus. *Acta Pharmaceutica Sinica B*, 2014, **4**, 301.

2. Wang M, Zhang G, Wang Y, et al. Design, synthesis and anti-influenza A virus activity of novel 2, 4-disubstituted quinazoline derivatives. *Bioorganic & Medicinal Chemistry Letters*, 2020, **30**, 127143.

10) The authors use the term "catalytic selectivity" in two very different contexts, either for the construction of O- vs. C- glycosides and to distinguish between acceptors. They should not use this term interchangeably.

Response #3-10: Thanks for the valuable suggestion. According to the suggestions, we have revised the “catalytic selectivity” to “substrate selectivity” in catalyzing different acceptors.

11) Finally, the authors have made use of the promiscuous selectivity of GTs to make potential new drugs. There are certain aspects here that are of interest and novel. But both the experimental approach (cf. the data missing on turnover to mono- vs. diglycosylated products) and the application seem to be limited. To the latter point, the authors have shown application of their compounds, but have not mentioned what the difference (or, potentially, the advantage) would be to marketed drugs such as the FDA-approved Empagliflozin. Furthermore, they have not addressed that there are other SGLTs - would there be side reactions pertaining to inhibition of these?

Response #3-11: Thank so much for this reviewer’s expertise. In the experiments of investigating SGLT2 inhibitory activity of C-glycosides, dapagliflozin, one of the FDA approved marketed drugs, was used as a positive control. As the lead compounds, the activities of our C-glycosides *in vivo* and *in vitro* were indeed lower than those of dapagliflozin. And their structures need to be further optimized according to their bioactivities. As we know, one of the most important side reactions of SGLT2 inhibitors is reproductive system infection. The infection is usually caused by bacteria, fungi and/or viruses due to the increasing urinary glucose excretion. According to our results, besides the SGLT2 inhibitory activity, our C-glycosides also exhibited other bioactivities due to the group B with different structures (**Figure r12**). For example, besides the SGLT2 inhibitory activity, **44a** exhibited anti-influenza A (H1N1) activity ($IC_{50} \sim 10^{-6}$ M, Table S6, SI). That indicated that changing the structure of group B

could bring antiviral activity, which might be helpful to reduce the incidence of reproductive system infection induced by SGLT2 inhibitors. Therefore, further structural optimization with group B of these lead compounds will be performed to obtain C-glycosides with antibacterial, antiviral or antifungal activity as well as better SGLT2 inhibitory activity.

Figure r12 Potential SGLT2 inhibitors synthesized by AbCGT

Above all, although these bioactive C-glycosides are not perfect SGLT2 inhibitors for now, these findings bring us a direction of applying CGTs in synthesis of bioactive compounds. We believe, with more and more CGTs with different catalytic properties being found/engineered and used in this application field, new SGLT2 inhibitors with better activity and little side effect will be discovered and economically synthesized.

Taken together, there are quite a few flaws in the manuscript that may not be solved by another round of revision, and may be addressed better by potentially publishing two separate papers on the synthetic and application aspects.

Response: Thanks for the helpful advice and good questions to our manuscript. And thanks a lot for pointing out the flaws and the missing detailed information in our manuscript. We have revised our manuscript carefully and thoroughly according to the reviewer's suggestions. After the revision of this round, we believe our manuscript has been greatly improved.

Thanks for the advice about publishing two separate papers on the synthetic and application aspects. This paper is made up by interrelated three parts, which include **Finding the new enzyme, Exploring the catalytic property** and **Applying the enzyme**. The three parts not only connect with each other but also exhibit a progressive relationship at the logical level. Therefore, we believe it is better to make the three parts as a whole to form a complete article.

1) Finding the new enzyme

Generally, up to now, more than 10 plant CGTs have been found. However, the discovered CGTs function primarily on *C*-glycosylating flavonoids, benzophenones, and other phenols with acyl groups including our previous work (Chen et al., *ACS Catal*, 2018). It's a new finding that AbCGT is able to efficiently *C*-glycosylate unnatural phenol scaffolds lacking acyl groups that known CGTs cannot catalyze. This feature of AbCGT might be important for its practical application in the discovery of new SGLT2 inhibitors.

2) Exploring the catalytic property

Although a lot of research has been done in exploring the substrate spectra of reported CGTs, the practical application of known CGTs is very limited. This work firstly provides a practical way for the application of CGTs in the synthesis of bioactive compounds for drug discovery. With more and more CGTs being identified, the application in practice should be advanced forward. We designed and synthesized diverse new potential SGLT2 inhibitors by the chemo-enzymatic approach. And some new *C*-glycosides with significant SGLT2 inhibitory activity both *in vitro* and *in vivo*, which can be further developed to be potential drug leads, were successfully synthesized.

3) Applying the enzyme

A whole-cell biocatalyst with the potential of industrial application in synthesis of target *C*-glycosides was constructed. The whole-cell biocatalyst can be reused more than ten times, but still remained high *C*-glycosylation activity. The recyclability and stability of the whole-cell biocatalyst could greatly decrease the cost of chemo-enzymatic approach for *C*-glycosylation. These findings bring possibility for CGTs to really progress from the laboratory to industry for the synthesis of clinical *C*-glycoside drugs and new biological *C*-glycosides.

Therefore, in this present work, we not only focused on finding new plant CGTs with different substrate scopes but also brought ideas about future practical application of the CGTs. **Finding the new enzyme**, **Exploring the catalytic property** and **Applying the enzyme** are a progressive relationship at the logical level and we believe it is better to make the three parts as a whole.

Thanks so much for the reviewer's comments.

Responses to Reviewers #4

Reviewer #4 (Remarks to the Author):

In this manuscript, the authors presented AbCGT is promiscuous GT, has the ability for C-, N-, S-, and O-glycosylation, accepts phenolic substrates without acyl group, and can use diverse sugar donors. The authors used *E. coli* as a whole-cell system for the chemoenzymatic synthesis of C-glycosides and some new glycosides demonstrate SGLT2 inhibition. It is an impressive finding however, there are some inconsistencies and phrasing that can be misleading.

We would like to thank this reviewer for his/her time to review our manuscript again, and his/her positive comments, and invaluable questions and suggestions which definitely and greatly improve our manuscript. We have carefully revised the writing mistakes and inappropriate phrasings. The point-by-point responses are listed as follows.

1) The authors refer to ref: 44 for homology modeling in the “Methods” section and Figure 6A. However, the text describes the homology model was based on UGT85H2 (lines: 326-328).

Response #4-1: Thanks so much for pointing out this mistake. In our first submission, the homology model was based on UGT85H2. However, during our resubmission, the crystal structure of GgCGT in complex with UDP/phloretin was reported in ref 44 (ref 32 in the present version). We found AbCGT showed a higher identity with GgCGT. What’s more, these two CGTs both exhibited high di-C-glucosylation activities. Therefore, we re-performed the homology modeling with the template of GgCGT in the current version of our manuscript, though the results of homology modeling with templates of UGT85H2 and CgCGT are similar. It is a clerical error that the statement was not revised in the manuscript (Lines: 326–328). So we have revised UGT85H2 to GgCGT in the present version of resubmitted manuscript (Lines 339–340) and we have also carefully checked out all the possible clerical errors about the homology model.

Lines 339–340 (in the new version): *“Homology modeling was performed with the template GgCGT (PDB entry, 6L5R.1), a di-C-glycosyltransferase.³²”*

2) Based on the ref:44, GgCGT is highly promiscuous C-GT, has the ability to C-, N-, S-, and O- glycosylation. While the introduction and conclusion section mention MiCGT and TcCGT1, no discussion or comparison with GgCGT has been made throughout the text.

Response #4-2: Thanks for the reminding and the helpful suggestion. Yes, GgCGT is a highly promiscuous CGT with di-C-glycosylation activity. We have added the information about the catalytic property of GgCGT in the **Introduction** and **Conclusion** sections. GgCGT was also added to the phylogenetic tree of CGTs in **Figure S1** of the supplementary information.

Lines 59–61 (in the new version): *“Recently, a highly promiscuous CGT (GgCGT) with C-, O-, N- and S-glycosylation activity was identified from Glycyrrhiza glabra.³² GgCGT could efficiently catalyze C-glycosylation of substrates containing a flopropione unit.”*

Lines 396–399 (in the new version): *“For examples, UGT708A6 from Zea mays exhibits both C- and O-glycosylation activities²⁴, MiCGT from M. indica showed C-, O- and N-glycosylation activities¹⁵, and GgCGT from G. glabra generates C-, O-, N- and S-glycosides.³²”*

3) While Figure 1A shows several sugar donors were tested with compound **17** and three sugar donors resulted in 100% conversion (one was almost 90%), the text does not give much description about the reason for using several sugar donors nor does it discuss the importance of the results, or their utility in synthesizing sugar derivatives of SGLT2 inhibitors. The sugar-donors study stands by itself without having much relevance to the rest of the text.

Response #4-3: Thanks a lot for pointing out this point. In **Figure 1**, we investigated the sugar donor promiscuity of AbCGT. AbCGT was able to accept UDP-Glc, UDP-Gal and UDP-Xyl with high conversion rates. As we didn't describe the reason why we selected the glucose rather than galactose and xylose as the sugar moiety of SGLT2 inhibitors in the following part of the manuscript, it indeed made this part misleading.

At first, given the donor promiscuity of AbCGT, we have also prepared the C-galactoside (**17b**) and C-xyloside (**17c**) besides the C-glucosides (mono-C-glucoside **17a** and di-C-glucoside **17aa**) when phloretin (**17**) was used as an

acceptor. According to the results of biological assays, the mono-*C*-glucoside **17a** showed the highest SGLT2 inhibitory activity among these four glycosides (**Table r1**). So in the following work, we selected glucose as the sugar moiety of inhibitors. We have added this part to the manuscript to make our statement logically clear and consistent. Thanks for the comments again.

Lines 193–200 (in the new version): “Therefore, besides the *C*-glucosides (**17a** and **17aa**), the *C*-galactoside (**17b**) and *C*-xyloside (**17c**) were also prepared and their structures were identified as phloretin-3'-*C*- β -*D*-galactoside (**17b**) and phloretin-3'-*C*- β -*D*-xyloside (**17c**) by MS and NMR, respectively (Figures S6, S7 and S68–S71, SI). Biological assays revealed that *C*-glucoside **17a** with only one glucosyl moiety exhibited higher SGLT2 inhibitory activity than those of other *C*-glycosides (**17aa**, **17b** and **17c**) (Table S5, SI). Therefore, UDP-Glc was selected as the sugar donor in the following application of AbCGT in synthesis of *C*-glycosides.”

Table r1. SGLT2 inhibitory activity of phloretin-3'-*C*-glycosides with different sugar moieties (Table S5, SI)

Glucosylated Products	Inhibitory Rates (%) ^a
17a	52.2
17aa	2.8
17b	–3.1
17c	1.7
Positive control	100

a. The inhibitory rates were obtained with *C*-glycosides under the final concentration of 10 μ M and with dapagliflozin (positive control) under the final concentration of 10 μ M.

4) The “chemoenzymatic” word is used both as hyphenated and non-hyphenated- please choose one.

Response #4-4: Thanks for the reminding. We have revised “chemoenzymatic” to “chemo-enzymatic” all through the manuscript.

5) Line 29: “the key sites” - this should be key residues.

Response #4-5: Thanks for pointing out this mistake for us. We have changed “key sites” it to “key residues” in the manuscript.

6) Lines 93-97 – this MS/MS fragmentation pattern requires a reference.

Response #4-6: Thanks for the suggestion. We have added the references (ref23, 36 and 37 in the new version) in Line 98.

23. Brazier-Hicks, M. *et al.* The C-glycosylation of flavonoids in cereals. *J. Biol. Chem.*, **284**, 17926–17934 (2009).

36. Kazuno, S., Yanagida, M., Shindo, N. & Murayama, K. Mass spectrometric identification and quantification of glycosyl flavonoids, including dihydrochalcones with neutral loss scan mode. *Anal. Biochem.*, **347**, 182–192 (2005).

37. Ferreres, F., Silva, B. M., Andrade, P. B., Seabra, R. M. & Ferreira, M. A. Approach to the study of C-glycosyl flavones by ion trap HPLC-PAD-ESI/MS/MS: application to seeds of quince (*Cydonia oblonga*). *Phytochem. Anal.*, **14**, 352–359 (2003).

7) It appears that Mn^{2+} and Mg^{2+} increase the yield by 50%- yet the enzyme reaction conditions mentioned did not use this any divalent metal ion- an explanation for not using divalent metal ion would be helpful.

Response #4-7: Thanks for pointing out this point and the helpful suggestion. In the part of **Effects of pH, temperature and divalent metal ions in enzymatic reactions**, the enzymatic reactions were performed with small amount (0.05 $\mu\text{g}/\mu\text{L}$) of enzyme AbCGT only for a short time (10 min) as long time incubation could give 100% conversion rates in different conditions. Yes, Mn^{2+} and Mg^{2+} can increase the yield by 50% in the AbCGT catalyzing reactions with condition of 30 °C for 10 min (Methods of “*Effects of pH, temperature and divalent metal ions in enzymatic reactions*”).

In the **Glycosyltransferase activity assays** and **Scale-up enzymatic reactions**, the enzyme is excessive and incubation time is 12 hours. Under these conditions, the conversion rates of reactions with or without Mn^{2+} both reached 100% after incubating for 12 h with 1 $\mu\text{g}/\mu\text{L}$ of AbCGT. The time courses of the AbCGT catalyzing reactions with Mn^{2+} or without Mn^{2+} are analyzed and shown in Figures r13 and r14, respectively. Although the divalent metal ion could speed up the reactions, it can't affect the final yields of the products with long time incubation.

Therefore, in the following **Glycosyltransferase activity assays** and **Scale-up enzymatic reactions**, divalent metal ions were not added. Thanks a lot for this helpful suggestion. We have added an explanation in the manuscript.

Lines 137–139 (in the new version): “*Mn²⁺, Mg²⁺, Ca²⁺ and Ba²⁺ greatly enhanced the catalytic activity of AbCGT (Figure S7C, SI). This enhanced catalytic activity was observed with a short reaction time (10 min) as longer time incubation resulted the same conversion rates.*”

Figure r13 HPLC analysis of reactions catalyzed by AbCGT at different time intervals with Mn²⁺. A) Initial reaction system; B) Reactions after incubating for 1 min; C) Reactions after incubating for 5 min; D) Reactions after incubating for 15 min; Substrate **46** was used as the acceptor and UDP-Glc was used as the sugar donor. MnCl₂ was added into the reactions with the final concentration of 5 mM. Reactions were performed at pH 7.4 and 30 °C with AbCGT (1 μg/μL). The yields of **46a** at different time intervals were shown.

Figure r14 HPLC analysis of reactions catalyzed by AbCGT at different time intervals without Mn²⁺. A) Initial reaction system; B) Reactions after incubating for 1 min; C) Reactions after incubating for 5 min; D) Reactions after incubating for 15 min; Substrate **46** was used as the acceptor and UDP-Glc was used as the sugar donor. MnCl₂ was added into the reactions with the final concentration of 5 mM. Reactions were performed at pH 7.4 and 30 °C with AbCGT (1μg/μL). The yields of **46a** at different time intervals were shown.

8) Lines 138-139: “albeit not for the native substrates” – in the absence of known substrates for the enzyme, this does not look appropriate.

Response #4-8: Thanks for the suggestion. We have deleted this statement “albeit not for the native substrates”.

9) Line 164: “high efficiency” – In enzyme biochemistry “efficiency” refers to Kcat/KM and not %conversion in 12 h assay.

Response #4-9: Thanks a lot for pointing out this for us. Yes, the “efficiency” refers

to $K_{\text{cat}}/K_{\text{m}}$ in enzyme biochemistry. According to the suggestion, we have revised the statement “high efficiency” to “high conversion rate”.

Lines 166–168 (in the new version): *“However, when the H-6 of **6** was replaced by a methyl group (electron-donating group) (**9**), AbCGT showed specific C-glycosylation activity with high conversion rate.”*

10) Line 171: “C-glycosylation site needs at least two hydroxyl groups in its ortho positions” – this is misleading as it sounds as if hydroxyl groups are needed on both sides of C-glycosylation site.

Response #4-10: Thanks for the reminding. We have revised the misleading statement about the relationship of C-glycosylation site and hydroxyl groups.

Lines 173–175 (in the new version): *“According to the structures of the substrates and isolated products, the C-glycosylation catalyzed by AbCGT needs at least two hydroxyl groups in the aromatic rings of acceptors and occurs in the ortho position of one phenolic hydroxy.”*

11) Lines 216-222: The description of the first mention of a compound should follow the ascending order of the compound’s number.

Response #4-11: Thanks for the suggestion. The description has been changed following the ascending order of the compound’s number.

Line 227-236 (in the new version): *“Firstly, three commercially available C-/N-substituted resorcinols (**27–29**) were prepared. Then we designed and chemically synthesized 25 potential substrates (**30–54**). Functional groups, including aliphatic carbon chains (**30–33**), saturated carbon cycles (**34–37**), carbon chain-substituted benzenes (**38–41**), and N- and S-heterocycles (**42–45**), were chemically attached to phloroglucinol by a one-step reaction (Figures S78–S107, SI).⁴³ What’s more, various derivatives (**46–54**) of 1-benzylbenzene-2,4,6-triol (**38**) and 1-phenethylbenzene-2,4,6-triol (**39**) were chemically synthesized to investigate the effects of different substituents in the B ring on the pharmacological activity (Figures S108–S125, SI).”*

12) Line 224: organize the compounds in ascending order of compound numbers when they are referred together.

Response #4-12: Thanks for the suggestion. We have organized the compounds in ascending order of compound numbers.

Line 237 (in the new version): *“Notably, out of all twenty-eight acceptors (27–54), twenty two (31–37 and 40–54) were novel compounds.”*

13) Line 250- order the compounds within the parenthesis as per the compound number

Response #4-13: Thanks for the suggestion. The order of “(44a, 49a, 50a, 46a and 47a)” has been revised to “(44a, 46a, 47a, 49a and 50a)”.

14) Line 325: “key amino acids”- change to “key amino acid residues”

Lines 330, 353: “key amino acid” -change to “key amino acid residue”

Response #4-14: The “key amino acid(s)” have been changed to “key amino acid residue(s)” in Lines 325, 330 and 353 as suggested.

15) Supporting Information- HPLC-Mass fragmentation pattern needed to identify 1a, 2a and 3a is almost invisible- please provide better data

Response #4-15: Thanks for the suggestion. We have provided better HPLC-MS data to clearly identify **1a**, **2a** and **3a** in the following figures r15–r17 (Figures S14–S16, SI).

Figure r15. HPLC-UV/ESI-MS analysis of AbCGT enzyme product using aglycon **1** and UDPG as substrates. A) HPLC-UV analysis of the AbCGT catalyzing reaction; B) Typical positive ion MS for the peak of **1a**; C) Typical negative MS^2 for peak of **1a**. (Figure S14, SI).

Figure r16. HPLC-UV/ESI-MS analysis of AbCGT enzyme product using aglycon **2** and UDPG as substrates. A) HPLC-UV analysis of the AbCGT catalyzing reaction; B) Typical positive ion MS for the peak of **2a**; C) Typical negative MS^2 for peak of **2a**. (Figure S15, SI).

Figure r17. HPLC-UV/ESI-MS analysis of AbCGT enzyme product using aglycon **3** and UDPG as substrates. A) HPLC-UV analysis of the AbCGT catalyzing reaction; B) Typical positive ion MS for the peak of **3a**; C) Typical negative MS^2 for peak of **3a**. (Figure S16, SI).

16) Supporting Information -HPLC-MS/MS fragmentation chromatograms are missing for compounds 9, 16, 19, 20, 21, 28, 30, 38, 39, 42, 43 and 44. These are needed even though NMR information has been provided.

Response #4-16: Thanks for the helpful suggestions. The HPLC-MS/MS fragmentation chromatograms for compounds **9**, **11**, **16**, **17**, **18**, **19**, **20**, **21**, **28**, **30**, **38**, **39**, **42**, **43** and **44** are shown in the figures r2–r6 of **Response #3-1** and the following figures r18–r27. We have provided these HPLC-MS/MS fragmentation chromatograms in the Supplementary Information (**Figures S2–S6**, S22, S30, S31, S38, S40, S48, S49 and S52–S54, SI).

Figure r18. HPLC-UV/ESI-MS analysis of AbCGT enzyme product using aglycon **9** and UDPG as substrates. A) HPLC-UV analysis of the AbCGT catalyzing reaction; B) Typical negative ion MS for the peak of **9a**; C) Typical negative MS² for peak of **9a**. (Figure S22, SI)

Figure r19. HPLC-UV/ESI-MS analysis of AbCGT enzyme product using aglycon **20** and UDPG as substrates. A) HPLC-UV analysis of the AbCGT catalyzing reaction; B) Typical negative ion MS for the peak of **20a**; C) Typical negative MS² for peak of **20a**. (Figure S30, SI)

Figure r20. HPLC-UV/ESI-MS analysis of AbCGT enzyme product using aglycon **21** and UDPG as substrates. A) HPLC-UV analysis of the AbCGT catalyzed reaction; B) Typical negative ion MS for the peak of **21a**; C) Typical negative MS² for peak of **21a**. (Figure S31, SI)

Figure r21. HPLC-UV/ESI-MS analysis of AbCGT enzyme product using aglycon **28** and UDPG as substrates. A) HPLC-UV analysis of the AbCGT catalyzing reaction; B) Typical negative ion MS for the peak of **28a**; C) Typical negative MS² for peak of **28a**. (Figure S38, SI)

Figure r22. HPLC-UV/ESI-MS analysis of AbCGT enzyme product using aglycon **30** and UDPG as substrates. A) HPLC-UV analysis of the AbCGT catalyzing reaction; B) Typical negative ion MS for the peak of **30a**; C) Typical negative MS² for peak of **30a**. (Figure S40, SI)

Figure r23. HPLC-UV/ESI-MS analysis of AbCGT enzyme product using aglycon **38** and UDPG as substrates. A) HPLC-UV analysis of the AbCGT catalyzing reaction; B) Typical negative ion MS for the peak of **38a**; C) Typical negative MS² for peak of **38a**. (Figure S48, SI)

Figure r24. HPLC-UV/ESI-MS analysis of AbCGT enzyme product using aglycon **39** and UDPG as substrates. A) HPLC-UV analysis of the AbCGT catalyzing reaction; B) Typical negative ion MS for the peak of **39a**; C) Typical negative MS² for peak of **39a**. (Figure S49, SI)

Figure r25. HPLC-UV/ESI-MS analysis of AbCGT enzyme product using aglycon **42** and UDPG as substrates. A) HPLC-UV analysis of the AbCGT catalyzing reaction; B) Typical negative ion MS for the peak of **42a**; C) Typical negative MS² for peak of **42a**. (Figure S52, SI)

Figure r26. HPLC-UV/ESI-MS analysis of AbCGT enzyme product using aglycon **43** and UDPG as substrates. A) HPLC-UV analysis of the AbCGT catalyzing reaction; B) Typical negative ion MS for the peak of **43a**; C) Typical negative MS² for peak of **43a**. (Figure S53, SI)

Figure r27. HPLC-UV/ESI-MS analysis of AbCGT enzyme product using aglycon **44** and UDPG as substrates. A) HPLC-UV analysis of the AbCGT catalyzing reaction; B) Typical negative ion MS for the peak of **44a**; C) Typical negative MS² for peak of **44a**. (Figure S54, SI)

17) Supporting Information - Page S24-S35- please order the data as per compound numbers

Response #4-17: Thanks for the suggestion. We have ordered the data as per compound numbers as suggested (Pages S25–S37 in the new version).

18) Supporting Information - Page S46-S77- please order the data as per compound numbers

Response #4-18: Thanks for the suggestion. We have ordered the data as per compound numbers as suggested (Pages S54–S95 in the new version).

19) Supporting Information - Pages S123 and S126- please order the data as per compound numbers

Response #4-19: Thanks for the suggestion. We have ordered the data as per compound numbers as suggested (Pages S131–S134 in the new version).

REVIEWERS' COMMENTS

Reviewer #3 (Remarks to the Author):

The authors have addressed all queries. I still think that this should be two or more separate papers, which is apparent by the referee's queries pertaining to many different aspects of the manuscript and the ensuing sheer amount of data that makes the work difficult to disentangle. But the work seems to be sound.

Reviewer #4 (Remarks to the Author):

I am satisfied with the revisions made by the authors. They have addressed the concerns and have improved the manuscript.